

**A-year Continuous Observations of Near-Surface Atmospheric Water**
**Vapor Stable Isotopes at Matara, Sri Lanka**
Yuqing Wu [1,2], Jing Gao [1,3,*], Aibin Zhao [1], Xiaowei Niu [1], Yigang Liu [1,]
[2], Disna Ratnasekera [4,5], Tilak Priyadarshana Gamage [6], Amarasinghe
Hewage Ruwan Samantha [6]
*1 State Key Laboratory of Tibetan Plateau Earth System, Resources and Environment,*
*Institute of Tibetan Plateau Research, Chinese Academy of Sciences, Beijing 100101,*
*China*
*2 University of Chinese Academy of Sciences, Beijing, 100049, China*
*3 Lanzhou University, Lanzhou 733000, China*
*4 China-Sri Lanka Joint Center for Education & Research, Guangzhou 510301, China*
*5 Department of Agricultural Biology, Faculty of Agriculture, University of Ruhuna,*
*Matara 81000, Sri Lanka*
*6 Faculty of Fisheries and Marine Sciences & Technology, University of Ruhuna,*
*Matara 81000, Sri Lanka*
* *Corresponding to*: Jing Gao (gaojing@itpcas.ac.cn)
**Abstract:**
Atmospheric water vapor stable isotopes are crucial for understanding
hydrological cycle processes under climate change. This study presents a year-long in-
situ monitoring of atmospheric water vapor stable isotopes ($\delta^{18}O$, $\delta D$) at Matara, Sri
Lanka, from March 2020 to February 2021 to assess how oceanic sources and moisture
transport influence coastal atmospheric moisture isotopic composition. We identified
clear seasonal patterns in the isotopic composition, with $\delta^{18}O$, $\delta D$, and d-excess
showing substantial variation between the southwest and northeast monsoon periods.
The primary moisture sources were the Arabian Sea and the Indian Ocean during the
southwest monsoon (May to September), characterized by depleted $\delta^{18}O$ from -20.4‰
to -9.1‰. During the northeast monsoon dominated period, the northern Bay of Bengal,





the Indian subcontinent, and Southeast Asia were primary moisture sources, displayed
enriched $\delta^{18}O$ (-23.9‰ to -7.5‰) and higher d-excess values (up to 25 ‰). The study
also identified significant influences of sea surface temperature and sea surface relative
humidity, on the isotopic composition of water vapor. Additionally, outgoing longwave
radiation (OLR) is a significant index used to gauge the intensity of convective activity.
Lower OLR values, indicative of stronger and deeper convection, were associated with
more depleted $\delta^{18}O$ in air masses. These findings help to improve the understanding of
influences of the monsoon and local meteorological condition on water vapor isotopes
in tropical region and provide new dataset on enhancing water vapor isotopic modeling
or atmospheric processes projection in coastal regions.
**Keywords:** Indian Summer Monsoon, Water Vapor Isotopes, Sea Surface Condition,
Convective Activity, Sri Lanka
**Short Summary**
This study monitored atmospheric water vapor isotopes for a year at Matara, Sri
Lanka. It found clear seasonal variations in $\delta^{18}O$, $\delta D$, and d-excess. There showed
depleted $\delta^{18}O$ during the southwest monsoon, while had enriched $\delta^{18}O$ and higher d-
excess during the northeast monsoon. Sea surface condition and regional convective
activity significantly influenced the isotopic compositions, improving understanding of
monsoon and local meteorological condition impacts on tropical water vapor.

## 1 Introduction

The Indian Summer Monsoon (ISM), occurring from June to September, is a
pivotal component of the Asian climate system, serving as the primary transport of
moisture from the Indian Ocean to the Indian subcontinent and the Tibetan Plateau (TP).
Monsoonal precipitation plays a crucial role in agriculture and water resources,
affecting the welfare of over 1.9 billion people in surrounding countries (Webster et al.,
1998; Goswami et al., 2016). The Tibetan climate and hydrology are profoundly
influenced by the ISM, as it contributes significantly to the regional water cycle by



delivering substantial rainfall during the summer months. This rainfall is essential for
maintaining the glaciers and permafrost in the TP, which are key sources of water for
many of Asia's largest rivers (Bookhagen and Burbank, 2010). The ISM's intensity and
variability can lead to significant fluctuations in water availability, affecting both
agriculture and hydropower generation in the region (Singh and Bengtsson, 2004; Gao
et al., 2014). Furthermore, the interaction between the ISM and the TP's topography
creates unique climatic conditions that influence weather patterns and extreme events
in the region (Liu and Chen, 2000).
The seasonal precipitation and its origins over the TP are inextricably linked to
the dynamics of the ISM (Dai et al., 2021). Previous studies have provided evidence
that isotopic records derived from precipitation over the TP offer insights into the
climatic fluctuations and distinct moisture attributes associated with the ISM (Gao et
al., 2013; Guo et al., 2017). The summer monsoon brings significant moisture from the
Indian Ocean, leading to substantial rainfall over the TP primarily during the monsoon
months during June-September (Yao et al., 2012). This seasonal influx of moisture is
critical for maintaining the regional hydrological balance and supporting the
ecosystems. Furthermore, the ISM's intensity and variability significantly influence the
interannual and decadal precipitation patterns over the TP, affecting the overall water
availability and climatic stability of the region (Kaushal et al., 2018).
The stable isotopic composition of river water (Bershaw et al., 2012; Li and
Garzione, 2017), precipitation (Rahul et al., 2016a; Cai et al., 2017), and water vapor
(Risi et al., 2008; Steen-Larsen et al., 2013b; Rahul et al., 2016b; Lekshmy et al., 2022)
serves as a valuable tool for understanding the origins and transmission processes of
atmospheric water vapor. Recent studies have significantly enhanced our understanding
of isotopic signals in convection regions, illuminating the complex interactions between
moist processes and isotopic compositions in tropical deep convection. In the winter
trades near Barbados, vertical transport and large-scale circulations have been identified
as primary drivers of isotopic variability at the cloud base, acting over timescales from
hours to days (Bailey et al., 2023; Villiger and Aemisegger, 2024). Investigations into



water vapor isotopes in the West African troposphere reveal that both convection and
mixing emphasize the important role of large-scale atmospheric circulation processes
in the variations of water vapor isotopes (Diekmann et al., 2021; de Vries et al., 2022).
The mechanisms by which convective activity lowers stable isotope values of water
vapor and precipitation are still under debate. Some researchers emphasize the
significance of condensation levels (Cai and Tian, 2016; Permana et al., 2016;
Thompson et al., 2017), while others point to raindrop re-evaporation and raindrop-
vapor isotope exchange during strong convection as crucial factors (Galewsky et al.,
2016). Additionally, unsaturated or mesoscale descending airflows that transport vapor
depleted in heavy isotopes to the lower atmosphere also contribute to lower isotope
values (Risi et al., 2008; Kurita, 2013). The influence of these processes varies with the
intensity of convective activity. These studies provide valuable insights. However, there
is a paucity of study on the Indian Ocean, particularly in relation to Sri Lanka. This gap
underscores the need to explore isotopic signals in this region, with reference to
established findings by Risi et al. (2008) and other seminal works. Comparison with the
above results, recent studies on water stable isotopes in the South Indian Ocean and
South Asian region have uncovered connections between local processes and
atmospheric circulation, shedding light on sea-surface dynamics (Midhun et al., 2013;
Rahul et al., 2016b; Bonne et al., 2019). Fractionation occurs during various phase
transitions, such as sea surface evaporation, condensation beneath clouds, re-
evaporation of raindrops, and diffusive exchange between water vapor and raindrops
(Stewart, 1975; Benetti et al., 2018; Graf et al., 2019). The occurrence of fractionation
unveils investigable spatiotemporal distribution patterns in the water isotopic
composition, encompassing water vapor and precipitation. Deuterium excess (d-excess
$= \delta D - 8 \times \delta^{18}O$) is a useful parameter for studying kinetic fractionation effects
(Dansgaard, 1964). Compared to other water stable isotopes, such as those found in
precipitation and surface water, the monitoring of atmospheric water vapor isotopes is
not limited by season, weather, or location (Angert et al., 2008). This capability for full-
time and full-space observation allows for the avoidance of information loss during



sampling, thereby providing a more comprehensive, continuous insight into the
evolving processes of atmospheric water vapor transport from diverse sources and a
thorough understanding of isotope transformation processes within the water cycle.
Evaporation at the ocean surface constitutes a significant component of the global
water cycle and is pivotal in accurately modeling climate change. The primary objective
of research on water vapor stable isotopes in the marine boundary layer aims to
elucidate the processes and influencing factors of evaporation isotopes (Craig and
Gordon, 1965). The d-excess of evaporated water vapor is predominantly impacted by
dynamic fractionation associated with sea surface temperature (SST), the relative
humidity of the sea-surface air ($RH_{SST}$, calculated relative to the saturation vapor
pressure at SST), and wind speed (rough or smooth) (Benetti et al., 2015; Benetti et al.,
2018). Investigations into the water vapor stable isotopic composition within the marine
boundary layer have been principally concentrated around regions including a large part
of the North Atlantic Ocean (such as Greenland, Iceland, Bermuda) (Steen-Larsen et
al., 2013a; Bonne et al., 2014; Benetti et al., 2018; Bonne et al., 2019), Bay of Bengal
(BoB) (Lekshmy et al., 2022), and the ocean throughout the Atlantic and Arctic Oceans
(Kurita, 2011). These studies have validated the negative relationship between d-excess
and $RH_{SST}$ (Uemura et al., 2008; Steen-Larsen et al., 2015), suggesting that wind speed
and SST exert limited influence on this correlation (Benetti et al., 2015). Observations
from the North Atlantic bolster this theory (Benetti et al., 2014). In addition, it also
highlights the significant variations in d-excess values from different moisture sources
(Kurita, 2011; Steen-Larsen et al., 2013b; Delattre et al., 2015). Subsequently, Benetti
et al. (2015) introduced a multi-layer mixing model, which is expected to advance the
accuracy of d-excess and water vapor isotope simulations. Due to the impact of dynamic
fractionation on sea surface evaporation, some studies have focused on simulating
observed d-excess under the closure assumption (Bonne et al., 2019). Furthermore,
researchers have used isotope atmospheric circulation models to assess mixing and
transport processes within the marine boundary layer (Benetti et al., 2015). Owing to
the minor influence of transport-induced fractionation, d-excess of the marine boundary



layer is conventionally employed to deduce moisture sources (Benetti et al., 2018).

Amidst the current backdrop of global climate change, observing stable isotopes

in atmospheric water vapor is vital for monitoring and comprehending climate shifts in
tropical low-latitude areas (Rahul et al., 2016b). Such research is instrumental in
providing a deeper understanding of near-surface water vapor dynamics, pinpointing
vapor sources and transport routes, and differentiating the contributions of atmospheric
water vapor to the water cycle. Positioned in the northern expanse of the Indian Ocean,
Sri Lanka experiences pronounced impacts from both the southwest monsoon and the
northeast monsoon (Fig. 1a, b). It emerges as a prominent origin region for monsoonal
water vapor in the TP. Therefore, investigating the dynamics and variations of near-
surface atmospheric water vapor stable isotopes at coastal stations, pivotal for
monitoring monsoonal water vapor source regions, enhances our understanding of
precipitation processes in the Indian Ocean. Oceanic evaporation serves as the inaugural
stage in the global water cycle phase transition. The primary objective of researching
water vapor stable isotopes is to comprehend the processes and controlling factors of
water isotopic variations.

In this study, we conducted continuous observations of near-surface atmospheric

water vapor stable isotopes in Matara, Sri Lanka, from March 1, 2020, to February 28,
2021. Our goal is to understand the main variations in moisture sources and
transmission processes in tropical coastal regions, and to explore how sea surface
processes, convective activity, and local meteorological factors affect near-surface
atmospheric water vapor stable isotopes at a coastal station, across daily, monthly, and
seasonal (monsoonal) time scales. Section 2 gives an overview of the study site,
covering meteorological and water vapor observations, calibration protocols, and
analysis methods. In Section 3, we illustrate the variability of isotopic and
meteorological parameters, analyze moisture sources, assess the impact of sea surface
processes on water vapor isotopes, and explore the relationship between water vapor
isotopes, convective activity, and local meteorological observations.





## 2 Data and methods

### 2.1 Study site and meteorological data

Sri Lanka (between 6°N to 10°N and 79° to 82°E), the southernmost country of the Indian subcontinent, is a key region for identifying the moisture source of the south Asian summer monsoon (Ravisankar et al., 2015). Features a tropical climate, Sri Lanka experiences four distinct monsoon seasons annually: the northeast monsoon from December to February, the first inter-monsoon from March to April, the southwest monsoon from May to September, and the second inter-monsoon from October to November (Malmgren et al., 2003; Jayasena et al., 2008). Most of the precipitation in Sri Lanka comes from the southwest and northeast monsoon systems, accounting for over 70% of the total annual precipitation (Fig. 1c). Precipitation formation in Sri Lanka primarily relies on organized convection associated with the Intertropical Convergence Zone (ITCZ) and low-pressure systems (Gadgil, 2003), and the moisture that derives precipitation is primarily derived from the Indian Ocean and BoB (Bandara et al., 2022). The southwest monsoon transports moisture from the Indian Ocean to southwestern Sri Lanka (Fig. 1a, b), leading to increased rainfall in the southwestern region of Sri Lanka compared to the northeast (Bavadekar and Mooley, 1981). Similarly, the northeast monsoon carries water vapor from the BoB to the north and northeast of Sri Lanka, where it produces disproportionately high amounts of rainfall compared to the southwest of the country (Dhar and Rakhecha, 1983; Wang, 2006).

An automated weather station (AWS) was installed at the University of Ruhuna, Matara (located at 5.94°N, 80.57°E) on the southern coast of Sri Lanka. It collected real-time meteorological observations, including air temperature, precipitation, relative humidity, vapor pressure, wind speed, and wind direction, from March 1, 2020, to February 28, 2021. Meteorological data are compared with water vapor isotopic data measured during the same period. The annual average precipitation is 2085 mm, and the annual average air temperature is 27.58℃ based on the European Centre for Medium-Range Weather Forecasts (ECMWF, https://cds.climate.copernicus.eu/eu/) reanalysis dataset (ERA5) from 2000 to 2020 (Fig. 1c) (Hersbach et al., 2020).



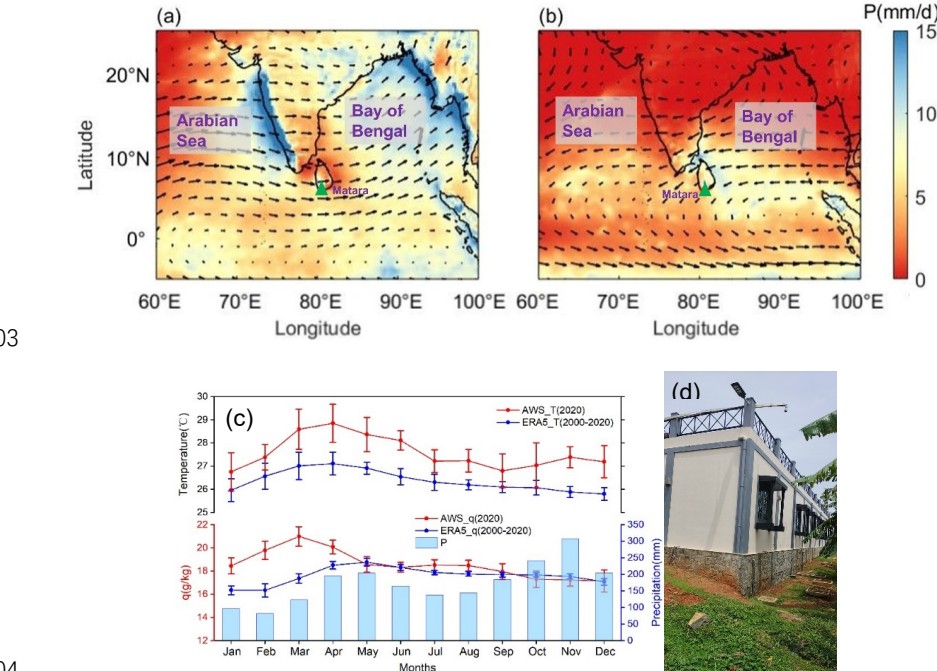

**Figure 1: Mean Wind Vectors (Arrows) at 850 hPa during the (a) 2020 Southwest Monsoon and (b) 2020/2021 Northeast Monsoon Seasons, along with Mean Precipitation (P, light blue rectangle) for the same. (c) Monthly Temperature and Specific Humidity (q) obtained from an automated weather station at Matara station (averaged for the years 2020-2021), as well as Monthly Average Temperature, Specific Humidity, and Precipitation (from ERA5, averaged for the years 2000-2020). (d) Photograph of the top floor platform at the University of Ruhuna where the system is installed.**

In this study, we used daily and monthly averages of outgoing longwave radiation (OLR, https://www.esrl.noaa.gov/psd/data/gridded/data.ncep.reanalysis.pressure.html) to quantify the convective activity. In addition, we used hourly data of 2m air temperature, 2m dew temperature, air pressure, precipitation, evaporation, SST, atmospheric boundary layer height (BLH), wind speed, and wind direction obtained from ERA5 for years 2000 to 2021, with a spatial resolution of 0.25°×0.25° and a temporal resolution of hourly. Studies have shown that ERA5 temperature, precipitation and other data provide good representations of the Matara equatorial climate and can



be used in lieu of missing observational data (Bandara et al., 2022). For the atmosphere
above open sea regions, $RH_{SST}$ is obtained by the following formula (Bonne et al., 2019):

$$RH_{SST} = RH_{2m\ air} \times \frac{q_{sat}(T_{2m\ air})}{q_{sat}(SST)} \tag{1}$$

where $RH_{2m\ air}$ is the relative humidity at 2m above the ocean surface, $q_{sat}(T_{2m\ air})$ is
the specific humidity at a saturated condition for a given 2m air temperature, and
$q_{sat}(SST)$ is calculated for seawater at salinity of 35 Practical salinity units (PSU)
(Curry and Webster, 1999).

The calculation formulas for air saturation specific humidity $q_{sat}(T_{air})$ and sea

surface saturation specific humidity $q_{sat}(SST)$ are:

$$q_{sat}(T_{air}) = \frac{0.622 \times E}{P} \tag{2}$$

$$q_{sat}(SST) = 0.98 \times q_s(\text{sea surface salinity of 35 PSU}) \tag{3}$$

among them, the calculation method of $q_s$(sea surface salinity of 35 PSU) is the same
as that of $q_{sat}(T_{air})$. E is the saturated water vapor pressure, obtained from the improved
Goff-Gratch formula (Goff and Gratch, 1946). P is atmospheric pressure, and the sea
surface pressure is taken as a fixed value of 1013.25 hPa for calculation.

## 2.2 In-situ Observation of Atmospheric Water Vapor Isotopic


## Compositions


At the Matara site, near-surface atmospheric water vapor isotope measurements

aim to establish a continuous, high-resolution dataset with one-second time intervals.
This study utilizes a Water Vapor Isotope Analyzer (manufactured by Los Gatos
Research (LGR) Inc.) in conjunction an LGR Water Vapor Isotope Standard Source
(WVISS model). The LGR instrument leverages Off-Axis Integrated Cavity Output
Spectroscopy (Off-Axis ICOS), a laser spectroscopic technique. This method integrates
a laser resonance cavity with a gas measurement chamber, where the laser oscillates
repeatedly between mirrors at the ends of the cavity. Only a small fraction of the laser



reaches the detector after traversing the sample gas thousands of times, effectively
increasing the chamber's thickness and significantly enhancing the water vapor
absorption signal. This allows for the detection of low concentrations of D and $^{18}$O in
water vapor (Liu et al., 2015). Compared to traditional methods, this spectroscopic
technique offers three advantages: it is compact and portable, enabling real-time field
monitoring; it can simultaneously measure $\delta^{18}$O and $\delta$D; and it has lower measurement
costs and requires less operator expertise, facilitating broader adoption.

The analytical system for measuring atmospheric water vapor stable isotopes in

Sri Lanka situated approximately 100 meters from the sea (5.94°N, 80.57°E, 10
meters), consists of four primary components: (1) Sampling inlet it positioned
approximately 5 meters above the ground, atop the office building of the China Sri
Lanka Joint Center for Education and Research at the University of Ruhuna (see Figure
1d). The inlet is equipped with a stainless-steel mesh to prevent the interference of
insects and directed downward to avoid direct rain splashes. (2) A 1/4-inch outer
diameter stainless steel tubing was used. The sampling tube is insulated with heating
tape and 2-cm thick insulation pipe to maintain warmth. (3) XX generates a constant
water vapor flow with known isotopic composition at different humidity levels. (4)
Water vapor isotope analyzer. In this study, the measurement precision of $\delta^{18}$O and $\delta$D
reaches 0.25‰ and 0.5‰, respectively, at a concentration of 2500 ppmv. This setup
minimizes external influences and maintains the integrity of the sampled water vapor.

The water vapor analytical system is located adjacent to the AWS, ensuring a high

level of synchrony between the water vapor stable isotope data and meteorological
measurements. We define wind directions ranging from 60° to 330°N are defined as
reflecting the ocean region, while those from 330° to 60°N reflect the land (Figure 1).

The $\delta$ notation, expressed in per mil (‰), is used to represent the atmospheric

water vapor stable isotopes, using the following equations:

$$R_{^{18}O} = \frac{^1H_2\ ^{18}O}{^1H_2\ ^{16}O} \qquad\qquad (4)$$



$$R_D = \frac{{}^1H\ {}^2H\ {}^{16}O}{{}^1H_2\ {}^{16}O} \qquad\qquad (5)$$

$$\delta_{sample} = \left( \frac{R_{sample}}{R_{VSMOW}} - 1 \right) \times 1000‰ \qquad\qquad (6)$$

Here, $\delta_{sample}$ represents either $\delta^{18}O$ or $\delta D$, indicating the $^{18}O$ or D isotope ratio
relative to Vienna Standard Mean Ocean Water (VSMOW) in the sample. $R_{sample}$ and
$R_{VSMOW}$ are the $^{18}O$ or D sample and VSMOW isotope ratios.

## 271   2.3 Calibration Protocol

In this study, we adhere to the calibration protocol proposed by Steen-Larsen et al.
(2013b). Briefly, the instrument calibration and data processing consist of three major
steps: (1) instrumental humidity-isotope response calibration, (2) Vienna Standard
Mean Ocean Water - Standard Light Antarctic Precipitation (VSMOW-SLAP)
calibration, and (3) drift correction (refer to Text 1 in the Supporting Information).
The water vapor concentration can influence the measured water vapor isotopic
composition, known as concentration- or humidity-isotope dependency
characterization. By introducing a constant stream of water vapor concentration with a
known isotopic composition at different humidity levels, we can establish the humidity-
isotope response function (Sturm and Knohl, 2010; Aemisegger et al., 2012). As this
function may vary over time, the humidity-isotope response calibration was repeated
monthly, using two standard samples with well-known isotopic compositions measured
at humidity levels ranging from 16,000 to 38,000 ppmv at intervals of 1000 ppmv, to
establish a correction function. Each measurement level was conducted for a minimum
of 25 minutes using the LGR WVISS. Our results are referenced to a humidity level of
20,000 ppmv. We compared our measurements to the international VSMOW-SLAP
scale, assuming a linear drift between calibration points.
All measurements are subject to instrumental internal drift, necessitating
correction through a specific drift-correction procedure. To compensate for this drift,
the LGR WVISS generates water vapor from a drift-standard bottle is measured for 25
minutes after each 12 hours of ambient air measurements. Furthermore, this drift-
standard water is sampled at each routine maintenance interval. Laboratory analyses of

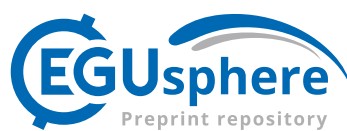

liquid isotopes have confirmed the stability of its isotopic composition over time. A
linear drift is assumed between each drift-standard measurement.

**2.4 Rayleigh Distillation Model and MBL-Mix Model**

The Rayleigh distillation model is employed to quantify isotopic variations during

phase changes (Dansgaard, 1964), wherein the residual air mass becomes drier with a
depletion in heavy isotopes following moist adiabatic vertical ascent (Gat, 1996):

$$R_r = R_0 f^{\alpha_v^l(T) - 1} \tag{8}$$

Here, $R_r$ and $R_0$ represent the isotopic ratio of residual vapor and initial vapor,

respectively. $\alpha_v^l(T)$ denotes the equilibrium fractionation factor, and f is the fraction
of residual water vapor.

By integrating the definition of isotope ratios as given in Equation (6), the

Rayleigh distillation model formula can be expressed in terms of isotopic content as
follows:

$$\delta_r = (\delta_0 + 1) f^{\alpha_v^l(T) - 1} - 1 \tag{9}$$

Where $\delta_r$ and $\delta_0$ are the isotope ratios relative to Vienna Standard Mean Ocean

Water (VSMOW) in the sample of residual vapor and initial vapor, respectively.

Meanwhile, we employ the mixing model to examine the isotopic characteristics

after the mixing of two air masses (Galewsky and Hurley, 2010):

$$R_{mix} = \frac{f[HDO]_1 + (1 - f) \times [HDO]_2}{f[H_2O]_1 + (1 - f) \times [H_2O]_2} \tag{10}$$

Where $R_{mix}$ represents the isotopic ratio of the mixed air mass, [HDO] and [$H_2O$]

denote the isotopic water vapor volume mixing ratio, and f is the mixing fraction.

Given that Matara is a coastal city, we utilize a framework employing water vapor

isotopes to study mixing processes in the marine boundary layer (MBL) (Benetti et al.,
2018), utilizing the following equation:

$$1 + \delta_e = \frac{1}{\alpha_k} \times \frac{\alpha_{eq}^{vl} \times (1 + \delta_{OC}) - RH_{SST} \times (1 + \delta_{MBL})}{1 - RH_{SST}} \tag{11}$$

Where $\alpha_{eq}^{vl}$ represents the equilibrium fractionation factor between vapor and

liquid, and $\alpha_k$ is the kinetic fractionation factor. $\delta_{OC}$ denotes the isotopic composition of





the ocean surface. We utilize $\alpha_{eq}^{vl}$ from Majoube (1971a, b) and $\alpha_k$ for the smooth
regime ($\alpha_k{}^{18}O = 1.006$ and $\alpha_k D = 1.0053$) (Merlivat and Jouzel, 1979).

## 2.5 Concentration-Weighted Trajectory and Moisture Source Diagnoses

To delineate water vapor transport paths and pinpoint moisture sources, we
employed the Hybrid Single-Particle Lagrange Integrated Trajectory (HYSPLIT)
model from the US National Oceanic and Atmospheric Administration (NOAA) to
compute backward trajectories of air masses associated with the southwest and
northeast monsoons. The Global Data Assimilation System (GDAS) with 1°×1° and 3-
hour spatial and temporal resolutions furnished the background meteorological data
from May 2020 to September 2020 and December 2020 to February 2021
(ftp://arlftp.arlhq.noaa.gov/archives/gdas1/). As atmospheric water vapor primarily
resides at altitudes below 2 km (Wallace and Hobbs, 2006), we initiated the backward
trajectories from a height of 50 m above the ground. Additionally, we computed 7-day
backward trajectories at 00:00h, 06:00h, 12:00h, and 18:00h during each monsoon
period and utilized K-means clustering to calculate specific humidity along each
trajectory.
Based on the HYSPLIT outcomes, we derived the concentration-weighted
trajectory (CWT) field at a resolution of 0.5°×0.5° (Hsu et al., 2003) using the in-situ
daily average $\delta^{18}O$ and d-excess in water vapor along each backward trajectory. This
facilitated the identification of potential moisture sources and assessment of
recirculation's influence on d-excess in water vapor (Salamalikis et al., 2015; Bedaso
and Wu, 2020; Xu et al., 2022). CWT ($C_{ij}$) was calculated as:

$$C_{ij} = \frac{\sum_{k=1}^{K} C_k \tau_{ijk}}{\sum_{k=1}^{K} \tau_{ijk}} \qquad (12)$$

Where (i, j) denote grid coordinates, k represents the trajectory index, K is the total
number of trajectories analyzed, $C_K$ is the concentration (here $\delta^{18}O$ and d-excess)
measured upon trajectory k's arrival, and $\tau_{ijk}$ is the residence time of trajectory k in grid



cell (i, j). During this computation, the residence time is substituted by the number of
trajectory endpoints in grid cell (i, j).

**3 Results**
**3.1 Seasonal Variability of Water Vapor Stable Isotope**

Figure 2 illustrates the hourly and daily averages of water vapor isotopes ($\delta^{18}$O,

$\delta$D, and d-excess) alongside temperature, relative humidity, atmospheric pressure, and
specific humidity from March 1, 2020, to February 28, 2021, at Matara station.

A clear seasonal cycle is evident in average values (Fig. 2 and Table 1) for relative

humidity, specific humidity, lifting condensation level (LCL), monthly precipitation,
and water vapor isotopic composition ($\delta^{18}$O, $\delta$D, and d-excess). Over the 12-month
observation period, average temperature and relative humidity stand at 27.6°C and
80.7%, respectively (Table 1). Temperature variations maintain consistent amplitudes
between monsoon and non-monsoon periods at around 10°C. Recorded minimum and
maximum temperatures are 22.3°C and 21.5°C, respectively. Specifically, comparing
monthly variations in air temperature and specific humidity (Fig. S3), both parameters
gradually decrease from relatively high values in May, reaching a minimum in
September, with monthly averages of 26.9°C and 18.5 g/kg, respectively. From January,
both air temperature and specific humidity show continuous increases, peaking in May
with monthly averages of 28.4°C and 21 g/kg. Mean relative humidity peaks in May at
95%, with lower values observed during winter and early spring (December to April),
reaching a minimum of 49.2% in January. From late May, specific humidity gradually
declines, stabilizing after mid-July and lasting until October, with levels ranging from
16 g/kg to 20 g/kg. During this period, significant oscillations of approximately 1.3 g/kg
occur during the southwest monsoon, with corresponding amplitudes doubled during
the northeast monsoon, at approximately 2.3 g/kg. During the southwest monsoon,
temperature, and specific humidity peak in May (monthly averages of 28.4 ± 1.4°C and
21.0 ± 1.1 g/kg). February marks the coldest and driest (specific humidity) month
(monthly averages of 27.4 ± 2.6°C and 17.1 ± 1.3 g/kg) during the northeast monsoon





(Fig. S3). The seasonal temperature variations exhibit modest amplitudes (Fig. 2),
attributed to the tropical location of the Matara station near the equator. Conversely,
relative humidity displays higher amplitude in seasonal variations compared to synoptic
variations. Furthermore, daily average SST consistently exceed the daily average 2m
air temperatures recorded by the AWS station (Fig. 2).
Yearly averages for water vapor isotopic values are -11.6‰ for $\delta^{18}$O, -79.5‰ for
$\delta$D, and 13.3‰ for d-excess, respectively, isotopic composition ranges from -23.9‰ to
-7.5‰ for $\delta^{18}$O, -173.2‰ to -53.4‰ for $\delta$D, and -1.2‰ to 28.1‰ for d-excess (Table
1). Monthly averages of water vapor isotopes ($\delta^{18}$O and d-excess) exhibit stability from
March to October, followed by sudden decreases. $\delta^{18}$O and $\delta$D show distinct seasonal
variations, with higher values during the southwest monsoon period and lower values
during the northeast monsoon period (Table 1). $\delta^{18}$O decreases through the southwest
monsoon, non-monsoon, and northeast monsoon periods, with mean values of -11.1‰,
-11.9‰, and -12.2‰, respectively. Extreme values of $\delta^{18}$O are observed during the
northeast monsoon, with a maximum of -7.5‰ and a minimum of -23.9‰. Conversely,
d-excess follows a reverse pattern to $\delta^{18}$O on both seasonal and monthly scales,
characterized by lower values during the southwest monsoon and higher values during
the non-monsoon period. d-excess increases sequentially through the northeast
monsoon, southwest monsoon, and non-monsoon periods, with mean values of 12.4‰,
13‰, and 14.7‰, respectively. The d-excess maximum occurs in November at 28.1‰
(monthly average of 15.2 ± 4.3‰), while the minimum of -1.2‰ is recorded in January
(monthly average of 11.3 ± 4.5‰). The d-excess peaks in April 2020 at 19.1‰,
indicating potential contributions from local recycling. Low specific humidity
corresponds to depleted $\delta^{18}$O and elevated d-excess values, indicating strong depletion
during long-distance transport from source regions to the observation station.

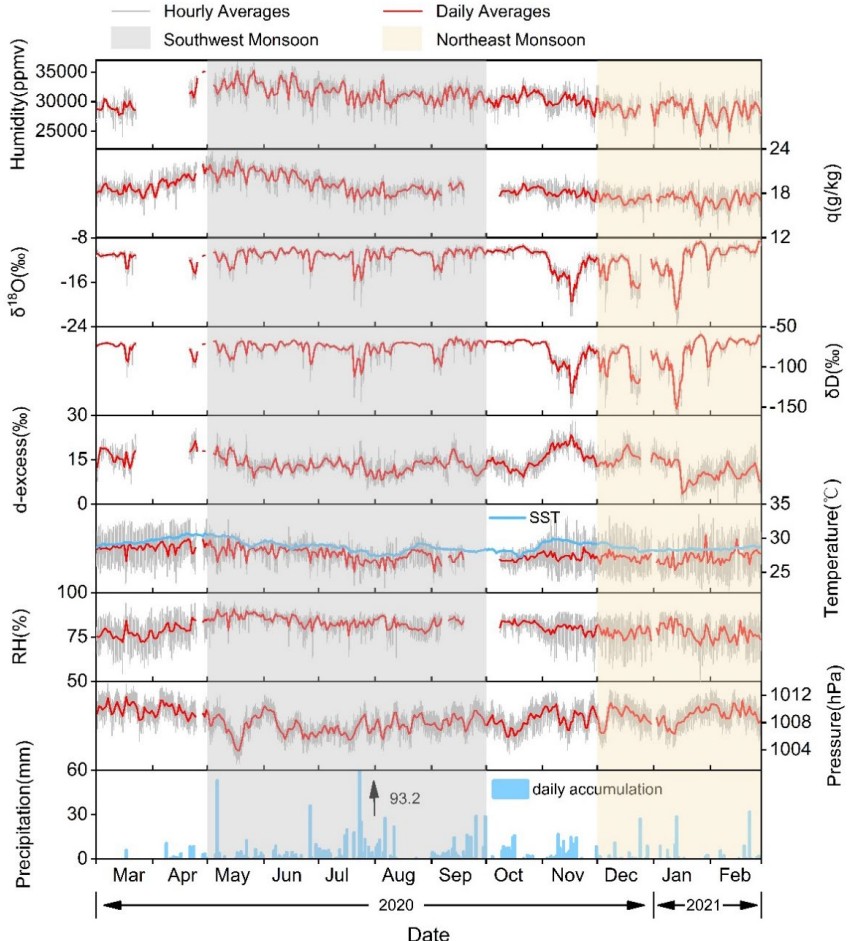

**Figure 2: Near-surface observations at the Matara station depict water vapor isotopes (δ¹⁸O, δD, and d-excess) alongside local meteorological parameters (humidity, specific humidity (q), temperature, relative humidity (RH), pressure, and precipitation) during non-monsoon, southwest monsoon, and northeast monsoon periods from March 1, 2020, to February 28, 2021. As Matara is a coastal city, local sea surface temperature (SST) is also plotted in blue.**



**Table 1: Summary of hourly-averaged data at Matara station during monsoon and non-monsoon periods from March 1, 2020, to February 28, 2021, including averages (bold), standard deviations (SD), minima, maxima, and the number of values (N) for δ¹⁸O, δD, d-excess, temperature (T), relative humidity (RH), specific humidity (q), atmospheric boundary layer height (BLH), and lifting condensation level (LCL). The maximum and minimum value for the year is all highlighted in bold italics.**

| Season | | $\delta^{18}O$ | $\delta D$ | d-excess | T | RH | q | BLH | LCL |
|---|---|---|---|---|---|---|---|---|---|
| | | (‰) | (‰) | (‰) | (°C) | (%) | (g/kg) | (m) | (m) |
| Non-monsoon | mean | **-11.9** | **-80.6** | **14.7** | **28.0** | **79.4** | **18.6** | **630.1** | **471.4** |
| | SD | 2.2 | 16.6 | 3.8 | 2.2 | 7.3 | 1.3 | 179.1 | 204.1 |
| | Max. | -9.0 | -65.3 | *28.1* | 33.2 | 94.2 | 23.0 | 1178.8 | 1283.1 |
| | Min. | -22.1 | -151.1 | 5.1 | 23.3 | 54.2 | 15.1 | *84.4* | 98.1 |
| | N | 1851 | 1851 | 1851 | 2617 | 2617 | 2617 | 2928 | 2617 |
| Southwest monsoon | mean | **-11.1** | **-75.7** | **13.0** | **27.6** | **83.8** | **19.4** | **741.4** | **348.7** |
| | SD | 1.3 | 9.6 | 2.8 | 1.5 | 4.5 | 1.5 | 149.0 | 118.4 |
| | Max. | -9.1 | -60.8 | 24.1 | 32.7 | *95.0* | *23.7* | *1564.4* | 938.9 |
| | Min. | -20.4 | -143.5 | 4.5 | 22.7 | 63.4 | 15.1 | 259.0 | *78.5* |
| | N | 3314 | 3314 | 3314 | 3192 | 3197 | 3192 | 3672 | 3192 |
| Northeast monsoon | mean | **-12.2** | **-85.1** | **12.4** | **27.1** | **77.4** | **17.2** | **516.4** | **524.7** |
| | SD | 3.0 | 22.0 | 4.29 | 2.4 | 7.8 | 1.2 | 139.4 | 224.0 |
| | Max. | *-7.5* | *-53.4* | 25.0 | *33.5* | 90.0 | 19.9 | 1125.7 | *1465.6* |
| | Min. | *-23.9* | *-173.2* | *-1.2* | 22.3 | *49.2* | *13.1* | 182.0 | 192.3 |
| | N | 1885 | 1885 | 1885 | 1993 | 1993 | 1993 | 2160 | 1993 |
| All | mean | **-11.6** | **-79.5** | **13.3** | **27.6** | **80.7** | **18.6** | **648.7** | **434.8** |
| | SD | 2.2 | 16.1 | 3.6 | 2.0 | 7.0 | 2.1 | 181.3 | 195.1 |
| | Max. | -7.5 | -53.4 | 28.1 | 33.5 | 95.0 | 23.7 | 1564.4 | 1465.6 |
| | Min. | -23.9 | -173.2 | -1.2 | 22.3 | 49.2 | 13.1 | 84.4 | 78.5 |
| | N | 7050 | 7050 | 7050 | 7802 | 7807 | 7807 | 8760 | 7802 |




For $\delta^{18}O$, $\delta D$, and d-excess, synoptic variations are also recorded (Fig. 2). Abrupt
changes occur in late July 2020 and from November 2020 to January 2021, associated
with synoptic events. Cumulative precipitation for July 2020 reached 451.8 mm, with
a notable rainfall event in late July recording daily rainfall of 93.2 mm. Isotopic $\delta^{18}O$
values emerged a sharp depletion from -10.4‰ to -20.4‰ within 20 hours during
isolated rainfall events. This depletion process of isotopes lasted for 6 days. Over a 75-
day period spanning from late southwest monsoon to mid-northeast monsoon,
noticeable fluctuations in isotopic $\delta$ values range from -22‰ to -11‰. during the
southwest monsoon from July 12 to August 7, $\delta^{18}O$ values varied from -20.4‰ to -
9.2‰, and $\delta D$ values ranged from -143.5‰ to -68.6‰. This finding is consistent with
water vapor isotopic $\delta^{18}O$ (-14.1‰ to -9.8‰) and $\delta D$ (-97.2‰ to 69.1‰) values
measured from July 12 to August 7, 2012, near the Bay of Bengal, although the local
minimum at Matara station is below the minimum in the Bay of Bengal (Midhun et al.,
2013). Stations such as Bangalore, Ponmudi, and Wayanad, all coastal like Matara,
exhibit water vapor isotopic values deficient in autumn and winter, mirroring
observations at Matara station (Table 2).
The atmospheric water vapor line serves as an indicator of the humidity conditions
at the vapor source and the fractionation processes along the transport path. The slope
reflects the extent of kinetic fractionation the vapor has experienced, while the intercept
indicates the humidity levels at the vapor source. Local Meteoric Water Line (LMWL)
for $\delta^{18}O$ and $\delta D$, compared with the Global Meteoric Water Line (GMWL), shows a
slope of < 8 during both monsoon periods (Fig. 3a). Seasonal variations are also visible
in $\delta^{18}O$ and $\delta D$ distribution patterns. Daily averages of water vapor isotopic $\delta^{18}O$ and
$\delta D$ demonstrate a strong correlation ($r = 0.96$) with a slope of 7.26 with a lower intercept
of 4.68. During the northeast monsoon, LMWL slope and intercept are higher compared
to other periods, indicating significant moisture recirculation. During The southwest
monsoon, lower slope (6.93) and intercept (1.18) are exhibited compared to other
periods, correlating with higher rainfall (Fig. 2).




**Table 2: Summary of observed water vapor isotope concentrations at various stations in India**
**and the Bay of Bengal, showing variations within each period.**

| Country or region | Station or location | Latitude (N°) | Longitude (E°) | Date | δ¹⁸O (‰) | δD (‰) | d-excess (‰) | References |
|---|---|---|---|---|---|---|---|---|
| India | Bangalore | 13.01 | 77.55 | Jun 1, 2012, to Sep 30, 2012 | -23.8 to -9.0 | -178.3 to -58.6 | -4.5 to 32.7 | (Rahul et al., 2016b) |
| | | | | Oct 1, 2012, to Feb 28, 2013 | -22.7 to -10.2 | -177.1 to -73.7 | -9.5 to 41.4 | |
| | Kolkata | 22.56 | 88.41 | May 3, 2019, to Oct 25, 2019 | -16.9 to -10.0 | -128.3 to -72.8 | -7.1 to 25.4 | (Bhattacharya et al., 2021) |
| | Roorkee | 29.87 | 77.88 | Feb 1, 2007, to May 31, 2007 | -17.0 to -3.0 | none | 32.0 to 70.0 | (Saranya et al., 2018) |
| | | | | Jun 1, 2007, to Sep 30, 2007 | -32.0 to -6.0 | | 40.0 to 87.0 | |
| | | | | Oct 1, 2007, to Dec 31, 2007 | -30.0 to -7.0 | | 30.0 to 60.0 | |
| | Ponmudi | 8.76 | 77.12 | Apr 1, 2012, to Nov 30 2012 | -24.1 to -8.6 | -170.0 to -51.0 | 6.3 to 26.5 | (Lekshmy et al., 2018) |
| | Wayanad | 11.51 | 76.02 | | -20.5 to -7.9 | -139.1 to -50.0 | 13.3 to 31.2 | |
| | Ahmedabad | 23.03 | 72.56 | Apr 1, 2007, to Apr 1, 2008 | -19.2 to -8.9 | -128.1 to -59.8 | 6.9 to 40.4 | (Srivastava et al., 2015) |
| | Chhota Shigri | 32.58 | 77.58 | none | -19.4 to -10.3 | -101.5 to -29.2 | 28.0 to 62.0 | (Ranjan et al., 2021) |
| Bay of Bengal | 6m | none | | Jul 1, 2012, to Aug 1, 2012 | -13.6 to -10.0 | -94.0 to -68.3 | 5.7 to 16.4 | (Midhun et al., 2013) |
| | 25m | | | | -14.1 to -9.8 | -97.2 to -69.1 | 6.9 to 19.4 | |
| | 25m | | | Nov 15, 2013, to Dec 1, 2013 | -19.9 to -11.0 | -136.6 to -69.4 | 13.3 to 31.0 | (Lekshmy et al., 2022) |

The observation period revealed a significant negative relationship between d-
excess and δ¹⁸O, where the rate of change for d-excess with δ¹⁸O is -0.68 ‰/‰ (r = -
0.55) (Fig. S4a), which is below the -1.4 ‰/‰ recorded at the southern Greenland
Ivittuut station and the -1.2~ -1.1 ‰/‰ range observed at NEEM station during the
summer (Steen-Larsen et al., 2013b; Bonne et al. 2014). Seasonally, the correlation



between the two variables weakens sequentially during the southwest monsoon period,
northeast monsoon period, and the non-monsoon period. The rates of change are -
0.94 ‰/‰ (r = -0.49), -0.69 ‰/‰ (r = -0.54), and -0.65 ‰/‰ (r = -0.44), respectively.
Similar patterns are detected for temperature–d-excess and specific humidity–d-excess
correlations. This pattern aligns with the incremental rise in the slope and intercept of
the water vapor line. Moreover, the concentrated distribution of vapor values during the
southwest monsoon and the highly scattered distribution during the northeast monsoon
are indicative of the corresponding seasonal distributions of the water vapor line.

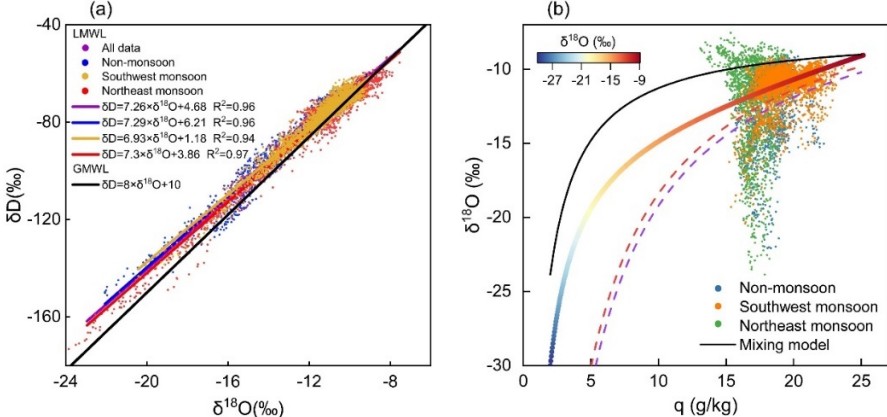


**Figure 3: (a) Co-variation of water vapor isotopic composition and meteorological parameters**
**during different monsoon and non-monsoon periods from March 1, 2020, to February 28, 2021.**
**The lines represent linear least-squares regressions (LMWL and GMWL) of δD (‰) as a**
**function of $\delta^{18}O$ (‰). (b) Scatter plot of observed hourly water vapor isotopic $\delta^{18}O$ vs. specific**
**humidity (q). The dotted red curve represents the Rayleigh distillation line during the**
**southwest monsoon. The dotted blue curve represents the Rayleigh distillation line during the**
**northeast monsoon. The solid black curve represents the mixing line. The colorful curve**
**represents the MBL-mix line.**
The q-$\delta^{18}O$ plots, combined with theoretical Rayleigh distillation curve, mixing
curve, and MBL-mix curve, were utilized to assess mixing conditions during the studied
periods (Fig. 3b). During the southwest monsoon, most measurements are clustered
between the Rayleigh curve and mixing curve, indicating isotopic variability dominated
by precipitation leaching process and moisture mixing process. Limited water vapor



measurements are scattered below the Rayleigh fractionation line, implying a
discernible impact of raindrop re-evaporation. Similarly, during the non-monsoon
period, most measurements are observed between the Rayleigh curve and mixing curve,
with a few located below the Rayleigh line. During the northeast monsoon, $\delta^{18}O$ spans
both upper and lower sides of the mixing curve and Rayleigh distillation curve. The
measurements substantially deviated from the Rayleigh curve and more depleted than
Rayleigh prediction, which is likely due to the influence of convective processes.

### 3.2 Diurnal Cycles

To evaluate diurnal cycles in isotopic composition and meteorological parameters,
we analyzed hourly averages at Matara station, particularly focusing on the pronounced
diurnal patterns during the northeast monsoon characterized by stable weather
conditions (low horizontal wind speed) (Fig. 4c-e).
All water vapor isotopic signals ($\delta^{18}O$, $\delta D$, and d-excess) and meteorological
parameters exhibit strong diurnal variations during both monsoon and non-monsoon
periods (Fig. 4). Overall, the diurnal variation of local meteorological factors reflects
the dynamic changes in the atmospheric boundary layer at Matara. During the daytime,
as solar radiation intensifies and the boundary layer develops, temperatures and wind
speeds increase from noon to afternoon, accompanied by a decrease in relative humidity
and led to significant evapotranspiration. At night, surface radiative cooling causes
temperatures to drop, resulting in near-surface calm conditions and gradual air
saturation, which points to a relatively stable atmospheric boundary layer. During the
southwest monsoon, $\delta^{18}O$, $\delta D$, relative humidity, wind speed, specific humidity, and
BLH are generally higher than the northeast monsoon and non-monsoon periods, while
d-excess and LCL are lower. In the early morning, $\delta^{18}O$ values steadily drop, reaching
their lowest level (-11.26‰) at around sunrise (~09:00 local time (LT)). Subsequently,
they increase throughout the day, peaking (-10.87‰) in the afternoon (~15:00 LT), with
a diurnal fluctuation of merely 0.45‰. Increased specific humidity between 10:00 LT
and 14:00 LT coincides with rises in air temperature and wind speed and a decline in
relative humidity (Fig. 4c-f). BLH peaks between 14:00 LT and 16:00 LT, slightly





delayed compared to other meteorological parameters. Conversely, the northeast
monsoon exhibits reversed diurnal variations for each parameter. During the northeast
monsoon, the daily variations of $\delta^{18}O$ and d-excess are significant, with the maximum
amplitude changes at 1.1‰ and 6.8‰, respectively. Specific humidity peaks from 10:00
LT to 16:00 LT, accompanied by increases in air temperature, wind speed, BLH, and
LCL. After 16:00 LT, specific humidity decreases alongside declines in isotopic δ
values and other meteorological parameters. The d-excess peaks (14.81‰) at 09:00 LT
and fluctuates until 23:00 LT, contrasting with the period from 04:00 LT to 09:00 LT
(Fig. 4b). The d-excess exhibits a W-shaped variability, reaching similar highs at 09:00
LT and 21:00 LT. Specific humidity exhibits a diurnal variation that aligns closely with
the $\delta^{18}O$ pattern, reaching its minimum before sunrise and peaking around midday
(10:00-15:00 LT). From the afternoon to evening, specific humidity stays relatively
high and stable. The diurnal variation during the southwest monsoon and northeast
monsoon periods is 1.28 g/kg and 2.32 g/kg, respectively. Similarities with Lena station
patterns (Bonne et al. 2020) suggest potential influences from moisture exchange
between the atmosphere and the ocean surface, particularly during the northeast
monsoon.

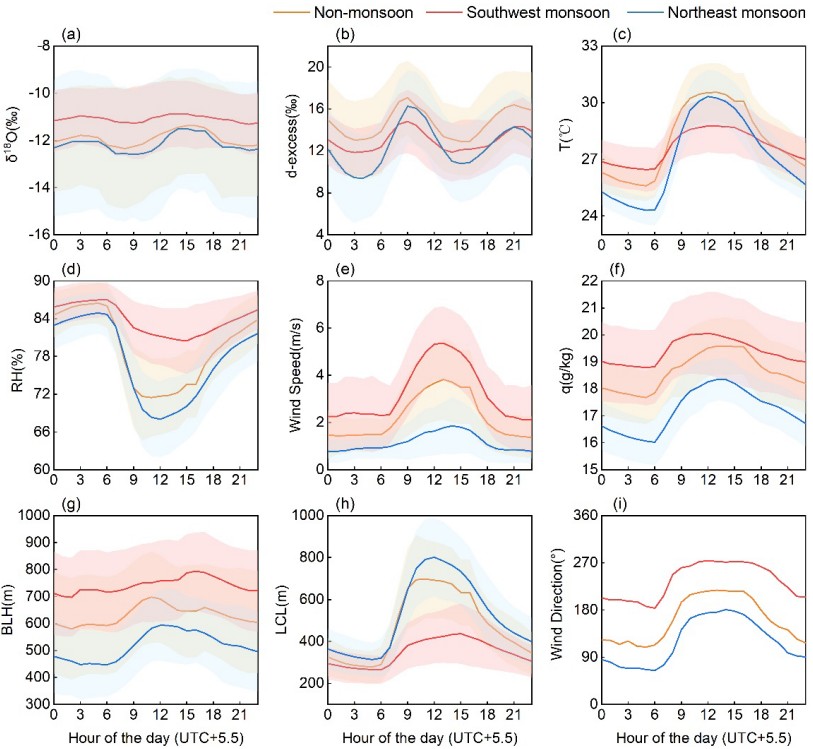

**Figure 4: Depicts average diurnal cycles of (a) δ¹⁸O, (b) d-excess, (c) temperature (T), (d) relative humidity (RH), (e) wind speed, (f) specific humidity (q), (g) atmospheric boundary layer height (BLH), (h) lifting condensation level (LCL), and (i) wind direction during the non-monsoon, southwest monsoon, and northeast monsoon periods. Shaded areas correspond to ±1 standard deviation.**

## 3.3 Sea Surface Evaporation Conditions in the Moisture Source Region

Understanding the processes and factors influencing water stable isotopic variations in ocean evaporation is crucial for exploring water vapor isotopic variations in the sea surface boundary layer. The primary determinant governing water vapor stable isotope shifts across different regions is the regional moisture transport process, characterized by differences in isotopic variations in the moisture source region, variations in meteorological conditions during evaporation processes, and divergences



of moisture transport pathways (Bonne et al., 2020). Thus, this section aims to reveal
essential factors driving the seasonal variation of near-surface atmospheric water vapor
stable isotopes at Matara, including water vapor origins, transmission routes, and sea
surface evaporation conditions in the source regions.
To further understand the different seasonal relationships between $\delta^{18}$O, d-excess,
and meteorological parameters, we analyzed potential seasonal differences in the main
moisture sources for water vapor transported to Matara Station during the 2020-2021
southwest monsoon and northeast monsoon by HYSPLIT. Trajectories during the
southwest monsoon and northeast monsoon show different origins of water vapor.
During the southwest monsoon, origins are mostly in the Arabian Sea (AS) and Indian
Ocean due to the northward movement of the warm South Equatorial Current, bringing
heavy rainfall to Matara. Conversely, during the northeast monsoon, most trajectories
originate in northeast India with lower specific humidity due to overland airflow, and a
small part from the BoB. The long transport distance results in more depletion of water
vapor isotopes at Matara station.
We calculated water vapor sources at Matara station for each month during the two
monsoon seasons. Fig. 5a shows that the primary moisture sources are the Indian Ocean
to the southwest and the BoB to the northeast of Matara. During the southwest monsoon,
water vapor predominantly originates from the Indian Ocean, encompassing wind
directions spanning 60° to 360°. Conversely, during the northeast monsoon, the primary
water vapor source shifts to the BoB, featuring wind directions from 0° to 225° and 330°
to 360° to exclude the influence of inland water vapor. Moisture from all sources shows
seasonal variations, with depleted $\delta^{18}$O values during the southwest monsoon and
enriched $\delta^{18}$O values during the northeast monsoon. The shift in water vapor source
from the AS to the southern Indian Ocean between May and September leads to
enriched water vapor $\delta^{18}$O values from August to September. Enhanced convective
activity and rainfall during the southwest monsoon result in $\delta^{18}$O depletion, while
tropical storms and hurricanes also contribute to $\delta^{18}$O depletion.

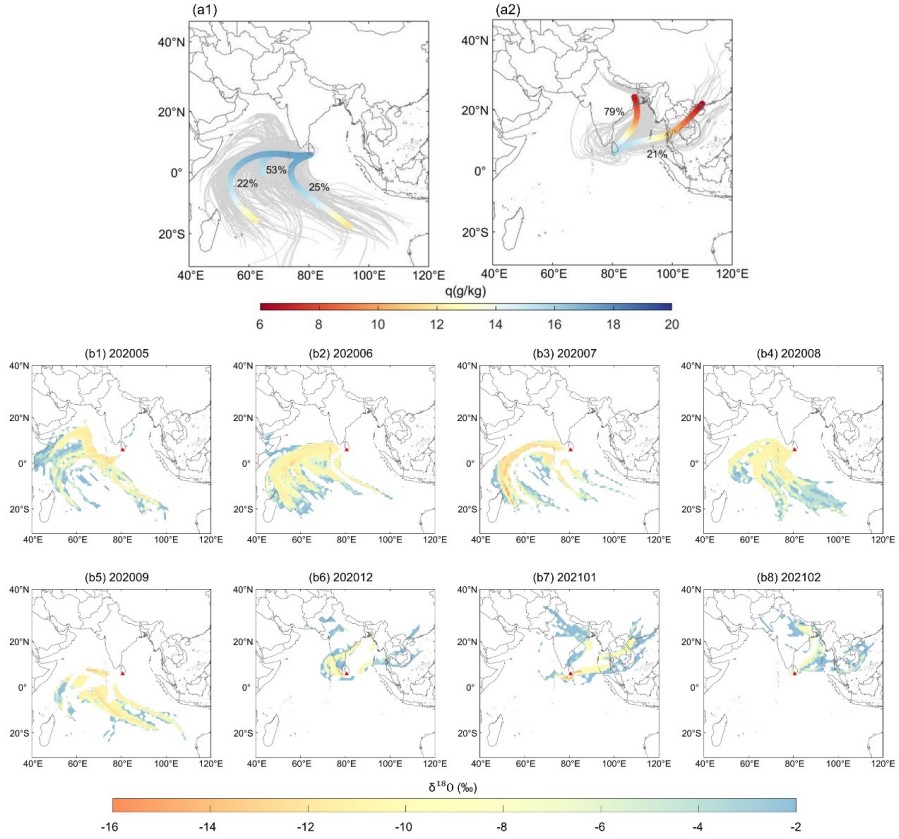

**Figure 5: Backward trajectories of water vapor tracks reaching Matara station (height of 50m) during (a) southwest monsoon and (b) northeast monsoon. The changes in specific humidity (q) along each clustered trajectory are shown in color. The black numbers indicate the percentages, reflecting the proportion of each clustered trajectory. Monthly concentration fields of water vapor isotopic δ¹⁸O for 168h HYSPLIT back trajectories during the two monsoons (b1-b8). Red triangle marks the study site.**

Similar seasonal variations are observed in d-excess values at Matara station, with lower values during the two monsoon seasons and higher values in the non-monsoon periods (refer to Table 2, Fig. 4). This seasonal variation in d-excess may stem from changes in relative humidity in the moisture source areas and further modifications during moisture transport.

Previous observational studies in the marine boundary layer have confirmed a



significant association between d-excess monitored at coastal observation stations and
$RH_{SST}$ in the proximate oceanic source areas (Pfahl and Wernli, 2009; Steen-Larsen et
al., 2015). During the southwest monsoon, $RH_{SST}$ values in "region a" ranged from 66%
to 84%, with SST fluctuating between 28.0°C and 30.6°C. During the northeast
monsoon, $RH_{SST}$ values in "region b" ranged from 54% to 84%, with SST fluctuating
between 28.1°C and 29.1°C. In comparison with the southwest monsoon, $RH_{SST}$
exhibits a comparatively drier tendency, accompanied by less pronounced variability in
SST. The rate of change in d-excess under the influence of $RH_{SST}$ in the BoB (during
the northeast monsoon) is -0.34 ‰/%. In comparison, the rate of change in d-excess
with the $RH_{SST}$ of the northern Indian Ocean (during the southwest monsoon) is -
0.51 ‰/%, suggesting that the evaporation from the northern Indian Ocean significantly
impacts local d-excess. Studies focused on the sea surface of BoB reveal that $RH_{SST}$
explains only 25% of the d-excess variation (d-excess = (-0.55 ± 0.14) × $RH_{SST}$ + (56
± 12); r = -0.5). The limited variation in relative humidity during the monsoon period
diminish the correlation, indicating that monsoon moisture plays a crucial role in the
isotopic composition of water vapor in the BoB (Midhun et al., 2013). Conversely, the
observed relationship between near-surface water vapor d-excess at Matara and relative
humidity in the surrounding oceanic region during the observational period, with
correlation coefficients of -0.61 and -0.62 (p<0.01), respectively (Fig. 6) reveals a
marked negative correlation between d-excess and relative humidity in the nearby
Indian Ocean and BoB, indicating that water vapor at Matara is predominantly supplied
by the adjacent marine environment. Notably, SST amplitude near the Matara station is
smaller than the near-surface air temperature, as depicted by the SST line in Fig. 2.




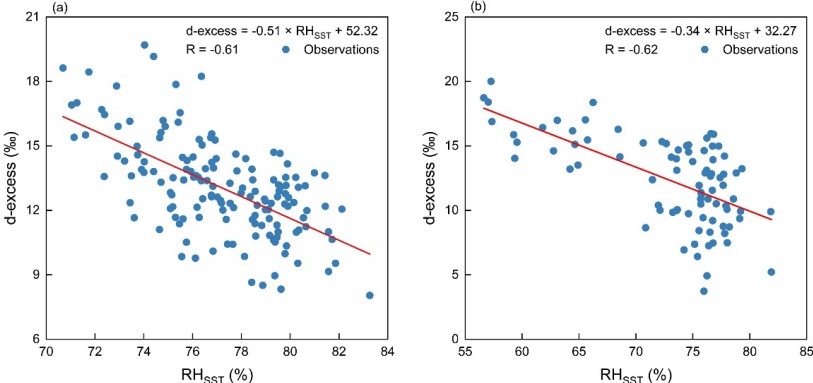

**Figure 6: illustrates the relationship between d-excess and $RH_{SST}$ during the (a) southwest monsoon and (b) northeast monsoon. Specific sea regions (Fig. S6) to the south (Region a: 3-6°N and 78-82°E) and east (Region b: 6-8°N and 82-85°E) of the observation station were selected to investigate the impact of sea surface meteorological conditions on near-surface water vapor isotopes during the two monsoon periods.**

## 3.4 Influence of Convective Activity

In the equatorial tropics, OLR mainly results from convective activity and cloud cover, which can impact the stable isotopic composition of precipitation (Ohring et al., 1984; Gao et al., 2013; Guo et al., 2017). Generally, higher OLR values are associated with weaker convective activity. Examining the correlation between stable isotopes of water vapor and OLR helps to understand the impact of convective activities along near-surface trajectories of water vapor stable isotopes at Matara station.

We calculate the spatiotemporal correlation of OLR, and precipitation amount with the measured water vapor isotopic compositions at Matara station, covering the period from March 2020 to February 2021. For each grid point in this region, we calculate the average precipitation amount by averaging over different numbers of days (n = 1, 2, up to 30) preceding each precipitation day. Lower OLR values represent the presence of deep convective clouds in this region, indicating relatively higher precipitation and associated with lower δ values.

Figure 7a represents the strong positive correlation (red regions) between rainfall and $\delta^{18}O$ during the southwest monsoon, mainly in the north BoB and India. This



correlation strengthens and extends over wider areas as n increases from 1 to 5.
Additionally, a strong negative correlation is evident in the northern Indian Ocean and
southern Arabian Sea, with correlations reaching a maximum for n = 2 days. During the
northeast monsoon, the spatial correlation distribution differs, with a negative
correlation observed in the southern Indian Ocean and BoB (Fig. 7b). Lower OLR
values in the Arabian Sea, the southern part of the BoB, and throughout Southeast Asia
correspond to a decrease in water vapor isotopic $\delta^{18}O$ at Matara station (Fig. 7c, d).
This pattern indicates that water vapor $\delta^{18}O$ during the northeast monsoon period is
influenced by convective activities in the Arabian Sea, South BoB, and Southeast Asian
regions. The stronger the convective activity, the more depleted water vapor isotopic
$\delta^{18}O$ the air reaching Matara becomes.

To examine the correlation between water vapor isotopic $\delta^{18}O$ and local

precipitation (Fig. 7e) and OLR (Fig. 7f), we choose a small region of 5°×5° with
Matara and calculate the time- and space- correlation for all grid points as described
above. The results show that the correlation with precipitation is negative during both
monsoon seasons as expected. The depletion of low-level water vapor $\delta^{18}O$ is related
to the transport and deposition of water vapor into the lower atmosphere through
convective activity (Kurita, 2013; Midhun et al., 2013; Lekshmy et al., 2014). The air
masses are re-supplied to the convective system through moisture recycling. This
results in a strong correlation between the isotopic composition of water vapor and the
convective activity of the previous day Fig. 7e and 7f. The residual water vapor is more
depleted in strong convective systems. In our study, the correlation reaches a high value
after about 5 days, indicating that the convective activity is sufficiently established to
affect the isotopic composition of water vapor. the correlation (for p < 0.05 and in
absolute terms) is indeed high for all n values, with maxima of about 0.48 for n = 3
days during the southwest monsoon and about 0.72 for n = 4 to 9 days during the
northeast monsoon.

The OLR correlation peaks at smaller time scales (refer to Fig. 7f), approximately

1-4 days, in contrast to precipitation, which peaks over larger time scales of 3-8 days.



We attribute this difference to the effect of cloud distribution on precipitation and OLR.
OLR has a stronger response to shallow clouds, while precipitation is more responsive
to both deep convective clouds and shallow clouds (Masunaga and Kummerow, 2006;
Schumacher, 2006). The OLR minimum occurs when thunderstorm clouds result in
more precipitation. Additionally, deep thunderstorm clouds, with short lifetimes and
consequently very low OLR (corresponding to highly depleted water vapor isotopic $\delta$),
exhibit a short memory effect on the correlation (peak occurs at smaller time scales)
(Gambheer and Bhat, 2000).

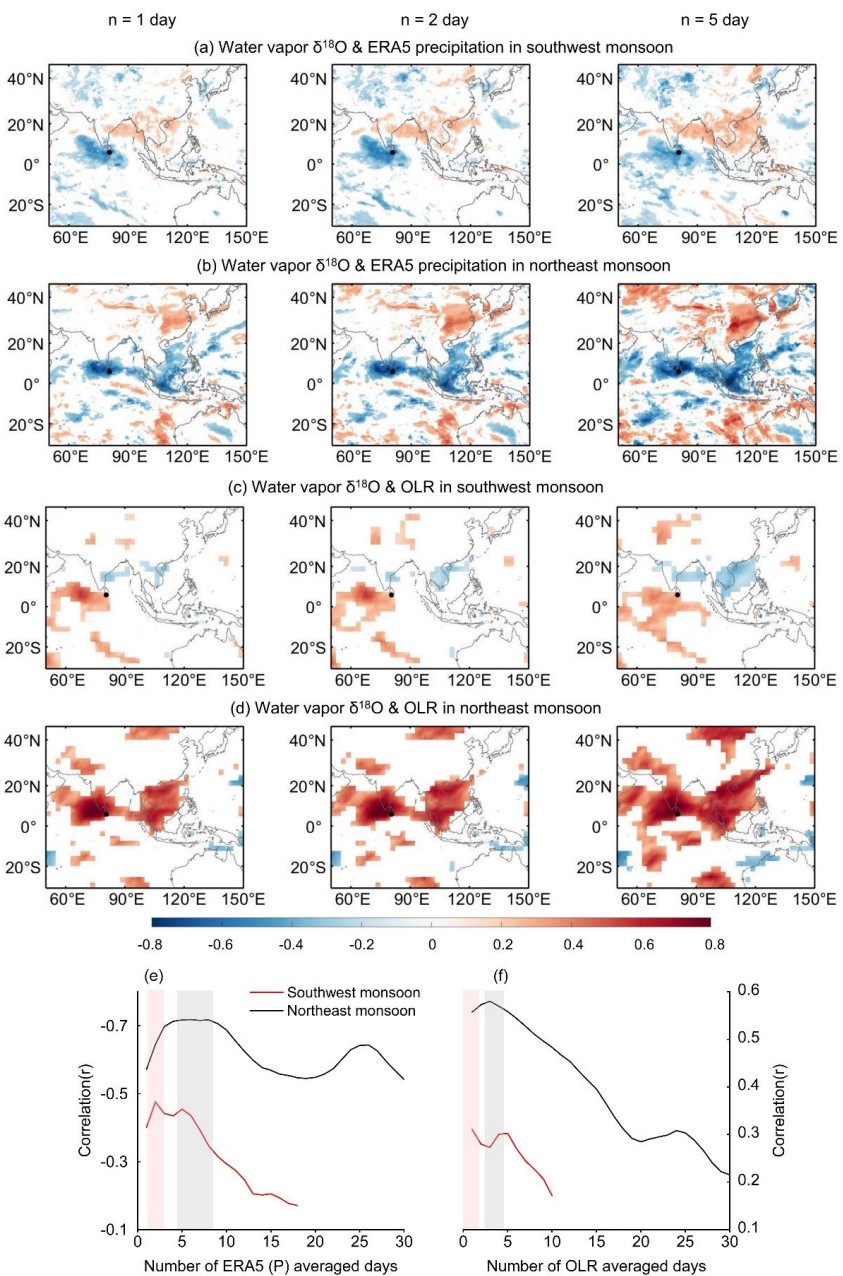

**Figure 7: Spatial correlation fields of water vapor isotopic composition and averaged ERA5 precipitation (P) during the (a) southwest monsoon and (b) northeast monsoon, along with the spatial correlation field of average outgoing longwave radiation (OLR) during the (c) southwest monsoon and (d) northeast monsoon. Averaging was conducted at each grid point**



**for three periods of n = 1, 2, and 5 days preceding each of the 153 days of the southwest**
**monsoon (a, c) and 90 days of the northeast monsoon (b, d). Correlation between δ¹⁸O and (e)**
**P, along with (f) OLR during the southwest monsoon (red line) and northeast monsoon (black**
**line) for values over n days (n = 1, 2, 3, … 30). Red (grey) area shows n-range with highest**
**correlation during southwest monsoon (northeast monsoon). Here, n represents the average**
**"moisture mixing time" of regional precipitation, affecting water vapor isotopes through the**
**transport of residual water vapor (Rahul et al., 2016b).**
During both the southwest and northeast monsoons, $\delta^{18}O$ exhibits weak correlation
with 2m air temperature (Fig. S7). Throughout the year, the relationship between 2m
air temperature and $\delta^{18}O$ in water vapor is $\delta^{18}O = 0.7T - 30.8$ (r = 0.32) (Fig. S9).
During the southwest and northeast monsoons, the relationships become $\delta^{18}O = 0.5T -$
24.95 (r = 0.39) and $\delta^{18}O = 1.46*T - 51.71$ (r = 0.43), respectively (Fig. S7). Daily
temperature and $\delta^{18}O$ values fluctuate less during the southwest monsoon than in the
northeast monsoon period (Fig. 4), possibly due to a weaker temperature inversion
during the southwest monsoon.
The correlation between $\delta^{18}O$ and relative humidity differs between the two
monsoon periods. During the southwest monsoon, $\delta^{18}O$ and relative humidity appear
uncorrelated (r = 0.01), consistent with previous findings (Rahul et al., 2016b).
Conversely, during the northeast monsoon, a robust negative correlation emerges
between $\delta^{18}O$ and relative humidity (r = -0.58). Similarly, the relationship between $\delta^{18}O$
and precipitation varies between both monsoon seasons (Fig. S7). During the southwest
monsoon, heavy precipitation leads to relatively high relative humidity and the
enrichment of heavier isotopes.

## 4. Summary and conclusions

This study presents one-year (March 2020 to February 2021) in-situ observations
of near-surface atmospheric water vapor isotopes ($\delta^{18}O$, $\delta D$) at Matara station in Sri
Lanka. These high-temporal resolution water vapor isotopic composition and
meteorological observations at Matara station provide a new sight to investigate the



water vapor isotopic dynamics from synoptic scale to seasonal scale. The variability of
water vapor isotopes at Matara station is influenced by local meteorological factors,
oceanic evaporation processes, and regional convective activities, depending on the
water sources and moisture transport. This dataset provides insights into multi-time-
scale variations in near-surface atmospheric water vapor in equatorial regions, and
provides information about the interactions between large-scale atmospheric moisture
transport and ocean evaporation.

Meteorological parameters exhibit diurnal variations during both monsoon and

non-monsoon periods. During the northeast monsoon, the diurnal fluctuation in $\delta^{18}$O,
temperature, and specific humidity are observed, with maximum values reaching 1.1‰,
6.0℃, and 2.3 g/kg, respectively. In contrast, variations of these parameters exhibit
small magnitude of 0.45‰, 2.3℃, and 1.3 g/kg during the southwest monsoon period.
Atmospheric temperature affects isotopic composition through its effect on isotope
fractionation. Additionally, a weak seasonal variability in near-surface water vapor
isotopes is observed, with $\delta^{18}$O typically showing high values (-11.1‰) during the
monsoon period and low values (-11.9‰) during the non-monsoon period. The d-
excess exhibits the lower value (12.7‰) during the monsoon period than that (14.7‰)
during the non-monsoon period.

The evaporation from the northern Indian Ocean significantly impacts local d-

excess. Contrary to previous research indicating a weak correlation (r = -0.5) between
d-excess in the Bay of Bengal and the sea surface relative humidity (RH$_{SST}$) (Midhun
et al., 2013), d-excess at Matara station exhibits a significantly negative correlation with
the RH$_{SST}$ during the monsoon periods, with the correlation of -0.61 and -0.62 (p<0.01)
in the northern Indian Ocean and the Bay of Bengal, respectively. This study
underscores the capability of near-surface d-excess to reflect the evaporation conditions
over these oceanic regions.

Consistent with previous research (Rahul et al., 2016b), large-scale rainfall and

regional convective activity (OLR) significantly impact isotope ratios at Matara station.
Notably, significant changes in $\delta^{18}$O are observed during a heavy rainfall event in July



2020, with a sharp decline in isotopic values from -10.4‰ to -20.4‰ within 20 hours.
During the southwest monsoon, strong cloud cover and high humidity over the ocean
may lead to $\delta^{18}O$ enrichment at the Matara station. The water vapor isotope
compositions observed during the southwest monsoon are similar as those observed in
the Bay of Bengal (Midhun et al., 2013). The deficiency of water vapor isotope values
at Matara station in autumn and winter is consistent with findings from other coastal
stations, such as Bangalore, Ponmudi, and Wayanad (Rahul et al., 2016b; Lekshmy et
al., 2018). Our results first pointed out that the correlation between OLR and $\delta^{18}O$ peaks
around 1-4 days, attributed to the impacts of cloud distribution.

This study contributes to a better understanding of the moisture origins at Matara

station and associated atmospheric transport. This combined water vapor isotope and
meteorological dataset offers extensive opportunities to further analysis of the typical
weather events, atmospheric patterns and ocean-atmosphere interactions in the
equatorial region. Ongoing observations of water vapor stable isotopes in this region
are strongly needed. This will support studies on interannual variability. Given the
anticipated numerous weather processes and hydrological changes in equatorial regions,
future research should explore the impacts of typical weather events, and ocean-
atmosphere interactions, deepening our understanding of extreme events and large-
scale atmospheric modes (e.g., ENSO, MJO, and IOD). Considering the temporal and
spatial variability in the interaction of tropical ocean-atmosphere systems, high-
resolution isotope model or satellite observation datasets should be employed for more
comprehensive analysis in the future.



**Acknowledgements:**

This work was funded by The Second Tibetan Plateau Scientific Expedition and Research (STEP) program (Grant No. 2019QZKK0208) and the National Natural Science Foundation of China (Grants 41988101-03 and 41922002), as well as the Innovation Program for Young Scholars of TPESER (QNCX2022ZD-01). We thank staff in the China Sri Lanka Joint Center for Education and Research, Mr. Charith Madusanka Widanage, and Dr. Di Dai for their invaluable support and assistance with measurements.

**Author Contributions:**

**Yuqing Wu:** Data curation, Formal analysis, Writing - Original draft preparation. **Jing Gao:** Data curation, Conceptualization, Methodology, Supervision, Writing - Review and Editing, Funding acquisition. **Aibin Zhao**: Writing - Review and Editing, Project administration. **Xiaowei Niu:** Data curation. **Yigang Liu:** Data curation. **Disna Ratnasekera:** Project administration. **Tilak Priyadarshana Gamage:** Project administration. **Amarasinghe Hewage Ruwan Samantha:** Data curation.

**Data availability:**

The ERA5 dataset is the latest reanalysis dataset published by the European Centre for Medium-Range Weather Forecasts (ECMWF) (Hersbach et al., 2020) (https://cds.climate.copernicus.eu/cdsapp#!/home). The Global Data Assimilation System (GDAS) published by the US National Oceanic and Atmospheric Administration (NOAA) (ftp://arlftp.arlhq.noaa.gov/archives/gdas1/). The water vapor isotopic compositions dataset will be available on the Zenodo research data repository after manuscript publication.

**Competing interests:**

The contact author has declared that none of the authors has any competing interests.



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
