# Peer review of "One-Year Continuous Observations of Near-Surface Atmospheric"

_EGUsphere, 2024_

## Author Comment (AC1)

The paper by Wu et al. (2024) presents the results of a year-long measurement campaign of stable water isotopes and atmospheric conditions in Matara, Sir Lanka. To explain the variability of d18O and d-excess they investigate the contribution from different moisture sources (trajectory analysis), the impact of near-surface relative humidity conditions of different regions, and the influence of convective activity. They conclude that the isotopic composition of water vapor varies seasonally and is influenced by the different moisture sources during the southwest (Arabian Sea) and the northeast monsoon (Bay of Bengal). Near-surface evaporation significantly affects local d-excess while convective activity is significantly correlated to variations in d18O.

The scientific significance of the paper is absolutely given, as continuous measurements of stable water isotopes in water vapor covering at least one year are highly needed. However, the scientific approach and the presentation of the results are sometimes chaotic and confusing and need to be improved. In the following, I will point out the main critical points and make some suggestions.

**## Scientific Quality/Approach**

**Classification into Monsoon Periods**

The main point is the distinction between the different monsoon periods. You aim to explain the stable water isotopic variability of water vapor in Matara, Sri Lanka. For your subsequent analysis you divided the time series into southwest, northeast, and non-monsoon periods. However, the isotopic signal and the variability of the isotopic composition within these defined seasons vary greatly. In particular, the non-monsoon period starts with a phase of high d18O values with a low d18O variability while the subsequent phase shows highly depleted d18O values with a strong d18O variability. A similar pattern can be observed during the northeast monsoon period with depleted values at the beginning and enriched ones towards the end. Note, the strongest depletion and enrichment are recorded during in one monsoon period, the northeast monsoon. As a result, the subsequent analysis and the respective interpretation are not convincing.

I suggest using a different classification depending on the d18O signal.

Reply: Thank you for your suggestion. We use the southwest monsoon, northeast monsoon, and non-monsoon periods for seasonal classification, mainly based on the literature review and understanding that Matara, as a coastal city, is influenced by both monsoons all year round. Therefore, our focus is on analyzing the changes in water vapor isotopes during different monsoon periods, paying particular attention to changes at different time scales, including daily, monthly, and seasonal. We have also added a separate discussion section on situations where isotopic changes are significant during monsoon or non-monsoon periods.

**Definition of Regions**

The definition of region a and region b (Line 570ff) is not defined in the text. Moreover, it is not clear, why you chose these regions.

Reply: Thank you for your suggestion. We have added the information of "Regions a and b" in lines 584-587.

Through theoretical models and observational research, it has been shown that dexcess in the oceanic boundary layer or at coastal observation points can effectively indicate the source of ocean evaporation water vapor. "Regions a and b" were chosen based on prevailing wind directions during the southwest and northeast monsoons (Section 3.3). During the southwest monsoon, water vapor mainly comes from the Indian Ocean, while during the northeast monsoon period, water vapor mainly comes from the Bay of Bengal. The region locations were chosen accordingly.

Similarly, it is not clear why you chose a  $5^{\circ}x5^{\circ}$  area for the spatiotemporal correlation (Line 625).

Reply: As mentioned in the manuscript (lines 641-644), we chose a  $5^{\circ} \times 5^{\circ}$  square box to minimize the effect of local variations.

**Used Variables**

The number of considered variables is enormous. However, it is sometimes unclear

where they come from (AWS, reanalysis etc.) and why you use all these variables as some are not relevant to your argumentation. For example, BLH and LCL are included in Table 1. Its label states that it contains the data from the station at Matara. In your data section BLH is from ERA5, which should be pointed out in the table and in the text. In contrast, the source for the LCL data is not given at all. Both variables are described in the text, but the results are not discussed in relation to the isotopic variability, neither in this paragraph nor elsewhere. Why do you present them here? Do you really need them?

Reply: Thank you for your suggestion. Following the reviewer's comments, we have added the calculation steps for LCL in Section 2.1 in lines 201-209 as new Equations 1-5. To some extent, LCL can reflect precipitation conditions. Therefore, we chose LCL for the analysis. Sections 3.1 and 3.2 are dedicated to a thorough discussion and analysis of BLH and LCL, allowing us to more comprehensively explore the factors influencing isotopic composition.

Moreover, the analysis is not consistent in its use of variables (see Table below), meaning I cannot discern a clear and coherent line of argumentation. You present BLH and LCL at the beginning, but you did not consider or mention it afterwards. Why is the consideration of wind speed and wind direction necessary and why is it only included in Fig. 4 but not in Fig 2? In Fig 6, you consider dexcess and RHsst although in the rest of the paper you try to understand the d18O variability. You calculated the spatiotemporal correlation between precipitation and d18O of water vapor, but it is not clear to me why you did consider precipitation at this point. At the end, you analyse OLR although it was not considered before. Why you did use the NCEP-NCAR reanalysis for OLR although you used ERA5 for the other variables. Note: it seems that OLR and therefore convective activity is very important at your

study site and agrees with the results of a recently published paper considering the tropical region of Ecuador, South America

• Landshuter, N., Aemisegger, F., & Mölg, T. (2024). Stable water isotope signals and their relation to stratiform and convective precipitation in the

tropical Andes. Journal of Geophysical Research: Atmospheres, 129, e2023JD040630. https://doi.org/10.1029/2023JD040630

I suggest including OLR at the beginning of the results section as time series (maybe at the end of Fig2). Additionally, you could consider generating spatial composites of OLR based on your defined periods.

|       | Hum
(ppmv) | δ18Ο | δD | Dexcess | Т | RH | Q | Press. | Precip | BLH | LCL | Wind |
|-------|---------------|------|----|---------|---|----|---|--------|--------|-----|-----|------|
|       |               |      |    |         |   |    |   |        |        |     |     | spe. |
| Fig 2 | х             | X    | х  | X       | x | х  | X | х      | x      |     |     |      |
| Tab 1 |               | X    | X  | x       | x | х  | x |        |        | X   | X   |      |
| Fig 3 |               | X    | x  | X       |   |    | X |        |        |     |     |      |
| Fig 4 |               | x    |    | X       | x | х  | X |        |        | x   | x   | x    |
| Fig 5 |               | x    |    |         |   |    | X |        |        |     |     |      |
| Fig 6 |               |      |    | X       |   |    |   |        |        |     |     |      |
| Fig 7 |               | X    |    |         |   |    |   |        | X      |     |     |      |
|       | 1             |      |    | - I.    |   |    |   | 1      |        |     |     |      |

Reply: Thank you for your suggestion.

1. See our response to the previous comment regarding BLH and LCL.

2. We considered wind speed and wind direction to better understand the monsoonal effects on Sri Lanka. Fig. S10 also explores the differences during the day- and nighttime periods, highlighting substantial diurnal variations in wind direction that directly impact water vapor stable isotopic composition, creating day-night differences.

3. Previous studies have shown that regional convective activity over a larger area can influence the water vapor stable isotopic composition. Thus, Section 3.4 focuses on how local convective activities, influenced by both the Indian Ocean and the Bay of Bengal, affect the water vapor isotopic composition at Matara station. Daily precipitation and outgoing longwave radiation (OLR) data were used to quantify the intensity of convection. Although the monsoon system is complex, most monsoon rainfall results from convective uplift, making rainfall at Matara a strong indicator for quantifying convection (Lekshmy et al., 2014).

Reference: Lekshmy, P.R., Midhun, M., Ramesh, R., and Jani, R.A.: 18O depletion in monsoon rain relates to large scale organized convection rather than the amount of rainfall, Sci. Rep., 4, 5661, https://doi.org/10.1038/srep05661, 2014.

4. Following the reviewer's comments, we have added the time series diagram of OLR as the new Fig. 2.

**Discussion and Interpretation**

All the results are only sporadically compared to results of other studies. However, even more important, the results are not discussed and interpreted together to receive an overall conclusion of the results. So, how do all your results fit together? Consider putting a discussion at the end of a paragraph or section or include a section "Discussion", which contains the discussion of your results with other studies and the interpretation.

Reply: Thank you for your suggestion. Following the reviewer's comments, we have added a new "Discussion" Section, which looks at seasonal versus synoptic variabilities and water vapor flux and comparative analysis of the main features and influencing factors. We also added two new figures in the *Supporting Information* (Fig. S11 and S12).

Line 580-583: Here the comparison to another study occurs before presenting your own results. Consider splitting and moving the information of this sentence: The explanation of the relevance of RHsst should be at the beginning of the paragraph or section. A comparison of values from another study should be stated after the presentation of your results.

Reply: Thank you for your suggestion. Following the reviewer's comments, we included an explanation of  $RH_{SST}$  at the start of the Section 3.3 (Lines 575-582). In the detailed analysis, we now first present the findings of our own study before making any comparisons with results from neighboring regions.

**## Presentation Quality**

The presentation of the study was often confusing for me. This is attributed to the overall structure of the results section and the use of an imprecise language within the paragraphs.

**Figure Selection and Order**

The overall structure was not easy to follow, which is linked to the already mentioned mix of analyzed variables, but also to the ordering of the figures and a missing explanation at the beginning of a section claiming why you did the respective analysis. For example:

-Figure 4: Why do you show the daily cycle of the variables? For presenting the new dataset? If so, consider switching Figure 3 and Figure 4 to keep the presentation of the new dataset together before going deeper into finding the explanation for the isotopic signal. Otherwise, I cannot see, why you included Figure 4. It does not contain new information that is already included in Figure 2 and is relevant for the subsequent argumentation/explanation.

Reply: Thank you for your suggestion. The intraday variations of isotopes and meteorological observation data shown in Fig. 4 are different from the time series presented in Fig. 2. Section 3.2 links the water vapor stable isotopes with local near surface water vapor, temperature, relative humidity, and precipitation. It serves for the analysis of how local meteorological elements affect the changes in stable isotopes of water vapor on a daily time scale, and allows us to focus on exploring the meteorological factors and influencing processes that occur in the local atmospheric boundary layer during the day and their potential impact on changes in  $\delta^{18}$ O and dexcess. In addition, we can explore how the diurnal variation in near surface atmospheric water vapor stable isotopes is influenced the differences in local diurnal water vapor condensation processes, which allows us to distinguish between the different impacts of local processes that affect the variation of near surface atmospheric water vapor stable isotopes in different seasons.

-Figure 6: Why do you consider d-excess here? It is hard to follow your argumentation at this point. Do you use d-excess to support the  $d^{18}O$  results? Or is it

an analysis of a different isotopic variable? In the first case, you should be more specific and explain the link and how the d-excess results support your results of the moisture source analysis. In the second case, consider putting Figure 6 at the end of your analysis.

Reply: Thank you for your suggestion. Ocean evaporation initiates the sequence of phase transformations that occur during the global water cycle and is a determining factor for isotopic variations. Understanding the controlling factors of oceanic evaporation is essential to elucidate isotopic shifts in the marine boundary layer. Observational data from numerous marine boundary layer studies have demonstrated a significant (typically negative) relationship between d-excess in near-surface water vapor and the relative humidity of the sea-surface air (RHSST). In cases where dynamic fractionation during air mass transport is either absent or minimal, d-excess can serve as an indicator of the moisture source region (Bonne et al., 2014). Our study utilized this method to examine sea surface evaporation conditions and found that the results are consistent with trajectory tracking, confirming that water vapor at Matara station is largely replenished via an influx from surrounding oceanic sources.

Reference: Bonne, J-L., Masson-Delmotte, V., Cattani, O., Delmotte, M., Risi, C., Sodemann, H., and Steen-Larsen, H.C.: The isotopic composition of water vapour and precipitation in Ivittuut, Southern Greenland, Atmos. Chem. Phys., 14, 4419-2014, https://doi.org/10.5194/acp-14-4419-2014, 2014.

-Note Figure 2 shows humidity (ppmv) and pressure, which are not considered in the text. Remove those variables from the plot or include them if they are relevant for your argumentation.

Reply: Thank you for your suggestion. Humidity (obtained from the LGR instrument) in shown in Fig. 2, and was utilized in the humidity correction process as detailed in Text S1.1 of the *Supporting Information*. Air pressure is employed to compute specific humidity and lifting condensation level (LCL). This is why we included time series plots for both variables.

**Imprecise Language and Structure within Paragraphs**

The language is often imprecise and does not only influence the readability of the manuscript but also the scientific quality. This mainly applies to the "Results" section. In the following, you find the most important cases.

**Introduction**

The introduction needs to be reconsidered. The first two paragraphs explain the influence of the monsoon on the Tibetan Plateau although the Tibetan Plateau is not considered in this study.

Reply: Thank you for your comments. Sri Lanka serves as an important hub for moisture transport from the Indian Ocean to the Indian subcontinent and the Tibetan Plateau. Therefore, understanding the atmospheric water vapor stable isotope composition and moisture sources in Sri Lanka's southernmost region of Matara, can provide insights into the variations in precipitation and water vapor influenced by the Indian summer monsoon over the Tibetan Plateau.

Line 105-112: This paragraph explains basic knowledge about stable isotopes and should be included before describing the results of recent studies (Line 81). Reply: Following the reviewer's comments, we have adjusted the order of the third

paragraph in the introduction (lines 82-94).

**Data and methods**

Line 201: Why did you use data from 2000-2020? Did you mean for the average annual precipitation and air-temperature? Is all the rest of the analysis restricted to the field campaign period?

Reply: We selected the period from 2000 to 2020 to better understand the multi-year average meteorological conditions of the study area (Fig. 1c). Here, 2m temperature, precipitation, and calculated specific humidity are used to reflect the long-term averages of temperature, precipitation, and specific humidity at Matara, i.e., to understand the region's average weather conditions. If we had used the same short period from 2020 to 2021 (for which we have observations), this would not

necessarily have been representative of the local climatic conditions.

Line 201ff: The structure is here not consistent, you present ERA5 data, then NCEP-NCAR and then you go back to ERA5 data. You say here that ERA5 is good for missing observational data. Did you use it for this purpose?

Reply: The meteorological reanalysis data used in the article (except for outgoing longwave radiation (OLR)) were obtained from ERA5 reanalysis dataset, and OLR dataset comes from NCEP reanalysis dataset. Through comparison and validation, ERA5 can be beneficial in supplementing missing meteorological data obtained *insitu* observation. Therefore, ERA5 is used to supplement the missing data observed.

Line 332: It is unclear how you derived the specific humidity of the trajectories Reply: Following the reviewer's suggestion, we have added the information of how specific humidity was derived (lines 337-341).

**Results**

Line 348: Starting a paragraph by describing what a figure shows is not very helpful for the reader. Instead, I would shortly explain the structure of the section.

Line 348-350: Figure 2 does also show SST, precipitation, and humidity, which is not stated. Are all variables from the weather station? If not, please make it clear and include the source of the dataset in the text (and in the Data section).

Reply: Following the reviewer's suggestions, we added the information on SST and local meteorological parameters in line 425. SST data come from the ERA5 reanalysis dataset (line 228). The remaining data shown in the original Fig. 2 are from AWS measurements.

Line 351: What kind of "average" values? Hourly, daily, monthly, annual? The consideration of different time scales is interesting, however, as in the given line, it is not clear which time scale you are considering. This is often not clear in the following paragraphs. It was confusing for me. To increase the readability of the text, I suggest

being more precise regarding the time scale. Moreover, it would help to present all results of one timescale together or always in the same order.

Reply: In Fig. S3, the data are presented as hourly averages. Monthly averages are subsequently calculated from these hourly values, with error bars included to show the amount of variability. Following the reviewer's comments, we added some additional information regarding the averaging time scales in lines 124-125 of the *Supporting Information*.

Line 352: Lifting condensation level, where does the data come from? Be more specific and take care of the structure. First, consider AWS data and if necessary, consider variables from other datasets. However, in your case, LCL is not important for your argumentation as you do not mention it in the rest of your manuscript.

Reply: Lifting condensation level (LCL) is calculated from air temperature, relative humidity, and air pressure measured by the AWS. We have added the formulas used by the AWS internal algorithm to Section 2.1 (lines 201-210, and equations 1-5). LCL is included in this study due to its role in illustrating the water vapor condensation process.

Line 356: No definition exists of the non-monsoon period: how is it defined? In Section 2.1 you introduce four monsoon periods but in your analysis you only have three.

Reply: This was indeed ambiguous in the manuscript. The non-monsoon months are defined as March-April and October-November. In Section 2.1, we had referred to the non-monsoon periods as the first and second inter-monsoon period instead, which may have created confusion. Furthermore, for the analysis we combined both non-monsoon periods into a single period. We have tried to remove this ambiguity and make this clearer (lines 186-187).

Line 351-end of paragraph: You describe temperature, specific humidity, and relative humidity in an inconsistent way. Try to structure your paragraph either by presenting

each variable or by going through the different months/seasons and compare the variables with each other, but always keep the same order.

Reply: Following the reviewer's comments, we adjusted the order of the variables in lines 374-397.

Line 383-385: True, however, this does not fit to the time series and is related to the classification issue I mentioned before.

Reply: Following the reviewer's comments, we adjusted the time series of the variables in lines 405-407.

Line 358: In your text is no reference to Fig. S2. Overall, you should check that every figure is referred to in the text and that the reference has the correct number. Reply: Following the reviewer's comments, upon careful review of the figure reference, we confirm that Fig. S3 is the correct reference, which was noted in line 388.

Line 363: You introduced the seasons in your text as monsoon seasons. For more clarity, try to avoid using other seasonal conventions such as winter, spring etc. throughout your text.

Reply: Following the reviewer's comments, we have adjusted the seasonal description in lines 384-388 to enhance clarity and ensure a more accurate representation of the data.

Line: 379: You present the results of d18O, dD, and d-excess. Nice. However, at some point you should state that you will only consider d18O in the following.

Reply: Following the reviewer's comments, we added "Consequently, the subsequent analysis will concentrate on the variations in  $\delta^{18}$ O." as a transitional sentence to provide explanation and clarification in lines 404-406.

Line 394: State or explain (maybe already in the introduction) that high values of dexcess are related to moisture recycling.

Reply: Following the reviewer's comments, we added "The high values of d-excess are related to moisture recycling." in line 416.

Line 395-396: I cannot see it in the figure. Is it a result or an explanation/interpretation? In the latter case, please make it clear and use a reference e.g.,

Reply: The analysis in this part is based on the inferred relationships between  $\delta^{18}$ O, specific humidity, and d-excess presented in Fig. 2. The sentence is intended as a concluding remark.

- Graf, P., Wernli, H., Pfahl, S., and Sodemann, H.: A new interpretative framework for below-cloud effects on stable water isotopes in vapour and rain, Atmos. Chem. Phys., 19, 747-765, https://doi.org/10.5194/acp-19-747-2019, 2019.
- Aemisegger, F., Pfahl, S., Sodemann, H., Lehner, I., Seneviratne, S. I., & Wernli, H. (2014). Deuterium excess as a proxy for continental moisture recycling and plant transpiration. *Atmospheric Chemistry and Physics*, 14(8), 4029–4054. https://doi.org/10.5194/acp-14-4029-2014
- Aemisegger, F., Spiegel, J. K., Pfahl, S., Sodemann, H., Eugster, W., & Wernli, H. (2015). Isotope meteorology of cold front passages: A case study combining observations and modeling. *Geophysical Research Letters*, 42(13), 5652–5660. https://doi.org/10.1002/2015gl063988

Similarly, Line 536-541, it is not clear, whether the stated information belongs to results of the study (please add a reference to the respective figure) or interpretation/background information from another study (please add reference). Reply: Following the reviewer's comments, we added the reference to Fig. 5a (line 580). Line 529: please order the factors in the way they appear in the following text.

Line 542: Did you really calculate water vapor sources as for example done by Sodemann, H., C. Schwierz, and H. Wernli (2008), Interannual variability of Greenland winter precipitation sources: Lagrangian moisture diagnostic and North Atlantic Oscillation influence, J. Geophys. Res., 113, D03107, doi:10.1029/2007JD008503.

or did you calculate the mean water vapor content of the clustered trajectories, or did you calculate the mean water vapor change along the clustered trajectories?

Reply: In this study, we employed the HYSPLIT backward trajectory model to compute the water vapor trajectories that reach Matara station. The analysis focused on changes in water vapor along its path to the Matara, and the moisture sources. Although trajectory frequency and clustering were calculated, the contributions of individual moisture sources were not analyzed.

Line 549: It is unclear, what the  $d^{18}O$  of the trajectories (Figure 5) means. It is probably not the mean traced  $d^{18}O$  along a trajectory, but it is the calculated concentration-weighted  $d^{18}O$ . How should this measure be interpreted?

Reply: Identifying the moisture source is an important topic in hydrological research, and different studies employ different app roaches to identify the moisture source, e.g., through a separate analysis of meteorological and isotopic data. Here, we employed the Concentration Weighted Trajectory (CWT) method and combine it with water vapor d-excess to infer the moisture source. The advantage is that d-excess is minimally affected by environmental factors, making it a good indicator for moisture tracing. Additionally, the analysis results can be visualized to intuitively identify the potential moisture sources. Previous studies (Salamalikis et al., 2015) have shown the suitability of the CWT model for this purpose.

Reference: Salamalikis, V., Argiriou, A.A., and Dotsika, E.: Stable isotopic composition of atmospheric water vapor in Patras, Greece: A concentration weighted trajectory approach, Atmos. Res., 152, 93-104, https://doi.org/10.1016/j.atmosres.2014.02.021, 2015.

Line 550: Results do not agree with the results of section 3.4

Reply: Following the reviewer's comments, we modified the sentence as follows (lines 558-560).

Moisture from all sources shows seasonal variations, with  $\delta^{18}O$  values lower during the southwest monsoon than during the northeast monsoon.

I hope my comments help to improve your manuscript.

---

## Author Comment (AC3)

This manuscript presents a new record of stable water isotope measurements in water vapor from Matara, Sri Lanka. The isotope measurements are compared with meteorological measurements from a nearby station and ERA5 / NCEP reanalysis data from the surrounding region to identify the most important drivers of isotopic variability at the site. The authors find differences in isotopic signatures between the northeast and southwest monsoon seasons, which they attribute to different moisture source conditions and convective activity.

This dataset is very valuable, especially given the scarcity of isotope measurements in the region. The measurement protocol appears to be sound (but I am not an expert, I hope the other reviewer(s) can check this better), and the analysis is well done with some exceptions. My main concerns are the moisture source diagnosis (see major comment 1) and the structure, specifically in the introduction and the results (major comment 2).

Major comments:

1) The trajectory analysis needs more explanation. For example, you wrote that you did a K-means clustering to calculate specific humidity along the trajectories. Why? Do you mean you did K-means clustering of the trajectories, and then calculate specific humidity along the resulting clustered trajectories?

Reply: The HYSPLIT model, using GDAS1 reanalysis datasets, generates specific humidity outputs for each trajectory along its path. Following the reviewer's comments, we have rewritten this part, providing more detailed explanations of how the backward trajectories are computed (lines 342-350).

2) Also you cannot simply assume that the end point of the trajectory is the moisture source. This is a very qualitative picture and does not provide more information than what is already known (i.e. moisture comes from the northeast during the northeast monsoon and from the southwest during the southwest monsoon). To get a more quantitative picture, you would have to look at moisture uptakes along the trajectories. For example, you could use positive changes in specific humidity, or

locations where there is evaporation from the surface. There are several moisture source diagnostics that could do this, e.g. WaterSip (Sodemann et al., 2008), HAMSTER (Keune et al., 2022), UTrack (Tuinenburg & Staal, 2020). For all of these you will also need more trajectories to get a representative picture of the air masses. One trajectory every 6h is not enough. I would recommend to start trajectories from several heights and from different horizontal locations around the measurement site.

Reply: Thank you for your suggestion. Following the reviewer's comments, we conducted backward trajectory tracking from Matara station from 16 additional point: specifically, the corner points of a 0.2° × 0.2° rectangle centered on Matara and four vertical levels (50m, 500m, 1200m, and 2000m) (see lines 338-342) giving now 20 points in total. Also Figure 5 has been revised based on these new trajectories and clustering results.

[Figure]

3) The structure of the text could be improved, in particular the introduction and the results section.

Reply: Thank you for your suggestion. Following the reviewer's comments, we have modified the structure of the introduction and the results sections.

4) The introduction now goes back and forth between monsoon, different isotope processes, and Sri Lanka. I would suggest restructuring it as follows: Motivate why the Indian Summer Monsoon is important for the Asian climate system (same as now), without mentioning isotopes yet. Then introduce stable water isotopes and why they are useful for studying the water cycle. Try to focus only on processes that are relevant for the study. Then write that there are not many studies on isotopes in the Indian Ocean and in particular Sri Lanka. Then introduce the new dataset.

Reply: Thank you for your suggestion. Following the reviewer's comments, we have modified the structure of the introduction, moving the content on isotope research in the Indian Ocean, especially Sri Lanka, to the penultimate paragraph in lines 142-156.

5) The results section introduces many different figures and variables and it is not always clear why. I would suggest showing only figures/variables that are important for the story and lead to the conclusions. Also make sure to describe where you got variables from, if you show them.

Reply: Thank you for your suggestion. Following the reviewer's comments, we have modified the Results Section to only discuss variables with essential significance. Furthermore, we carefully examined the variables employed in the results section and included an explanation of the sources from which these variables were derived.

Minor comments

Title: "A-year continuous observations" is grammatically wrong. Change to "One-year continuous observations" or "One year of continuous observations".

Reply: Thank you for pointing this out. We have corrected the title to "One-year Continuous Observations of Near-Surface Atmospheric Water Vapor Stable Isotopes at Matara, Sri Lanka".

L27&30: This is a bit confusing, -20.4‰ to -9.1‰ does not seem more depleted than -23.9‰ to -7.5‰.

Reply: Following the reviewer's comments, we have adjusted the sentence (see lines 27-31).

L29: displayed -> characterized by

Reply: We have changed the word to "characterized by" as suggested (line 30).

L32: No comma after humidity

Reply: We have deleted the comma after humidity in line 32.

L35: The findings don't provide a new dataset (second part of the sentence), rather the other way around.

Reply: We have adjusted the sentence (lines 37-39).

L37: "in tropical regions and provide a new dataset for enhancing..."

Reply: See above reply.

L44: There -> They

Reply: We have rephrased this to "The results" to avoid any ambiguity (line 45).

L45: Again, not really (cf. L27&30).

Reply: We have rephrased the sentence (Lines 45-47).

L46: The sea surface condition does not improve the understanding.

Reply: We have adjusted the sentence to "sea surface evaporation" in Line 47.

L178: Features -> Featuring

Reply: Changed (line 180).

L182: "Most of the precipitation…". With 8 out of 12 months attributed to either southwest or northeast monsoon, 70% is actually not so high (only 3.3% more than the average precipitation amount).

Reply: Following the reviewer's comments, we re-examined the original data and calculated that the rainfall recorded by the Automated Weather Station at Matara during the southwest and northeast monsoons accounted for 78% of the annual precipitation. Therefore, we have adjusted the sentence to use this percentage rather than "most" (line 188).

L186: derives -> forms/produces

Reply: Changed (line 190).

L190: Similarly -> In contrast

Reply: Changed (line 194).

L198: "Meteorological data are compared …": I would move this sentence to the beginning of Section 2.2.

Reply: Done.

L202: Do you mean Fig. 1a&b?

Reply: We meant Fig. 1c. This has been added as a reference (line 217).

L207: an -> the

Reply: Changed (line 221).

L208: What does "averaged" mean here? I thought you only have one year.

Reply: We mean "monthly" averages. This has been added for clarity (line 221).

L213: Instead of the link, just cite NCEP? Does ERA5 not provide OLR?

Reply: Although ERA5 also provides an OLR dataset, we chose the OLR dataset from NCEP, as referenced by the link in line 226.

L214: You do not really use all of these ERA5 variables, do you? I would mention only those that are used (and relevant). Also why 2000 to 2021, why not only 2020-2021?

Reply: Following the reviewer's comments, we rechecked the variables we used from the ERA5 reanalysis data. We use 2m air temperature, 2m dew temperature, and air pressure to calculate specific humidity. For the wind vector plots (Fig. 1a, b), we used wind speed and wind direction at 850 hPa. Precipitation data was used as the background for Fig. 1(a, b) to illustrate the distribution of tropical precipitation. We used SST for comparisons with the temperature at Matara and to assess sea surface evaporation conditions. Additionally, we performed an analysis about the effect on water vapor stable isotopic composition using the atmospheric boundary layer height. The only unused variable was "evaporation", which has been deleted (line 200).

We selected the period from 2000 to 2020 to gain an understanding of the climatological averages at the study site. A one-year period would have been too short to be representative of local climatic conditions.

L218: hourly -> one hour

Reply: Changed (line 231).

Equation 3: This equation is not very clear. It looks like q_s is a function of sea surface salinity of 35 PSU, but what you mean (I assume) is that it is q_sat(SST) at a salinity of 0 PSU, while the left hand side is q_sat(SST) at a salinity of 35 PSU.

Reply: We have also added "sea surface salinity of 35 PSU" after "$q_{sat}$ (SST)" in Equation 3 and line 240.

L231: I think here it would make more sense to take the atmospheric pressure (same as for q_sat(T_air)), because it is probably not constant, and the difference in pressure

between 2m and the sea surface is negligible. Assuming a constant sea surface pressure might introduce artificial variations in RHsst.

Reply: Following the reviewer's comments, we have changed the pressure to atmospheric pressure.

L237: in conjunction with

Reply: Following the reviewer's comments, we have added the word "with" after "in conjunction" in Line 251.

L250: How far away from the AWS is the water isotope analyzer?

Reply: The distance is about 5 m. We have added this to the main text (line 249).

L250: is situated, is positioned, and consists

Reply: Has been corrected (Lines 264-265).

L251: Could you add the numbers describing the different components to Figure 1d?

Reply: Following the reviewer's comments, we have added the numbers describing the different components to Fig. 1d.

L257: What is XX?

Reply: This was a placeholder that was left in the text by mistake. We have replaced it with the corresponding text (lines 263-265) as follows:

"The calibration unit generates a constant water vapor flow with known isotopic composition at different humidity levels. "

Thank you for pointing this out.

L264: Remove "are defined"

Reply: Removed (line 279).

Equations 6 & 7: Actually, the R values are the ratios of the isotopes rather than the

isotopologues (Coplen, 1994), i.e. R_18O = 18O / 16O and R_D = D / H

Reply: We have corrected Equations 9 & 10 accordingly. Thank you for pointing this out.

L270: Add "respectively" at the end.

Reply: Done (line 285).

L291: This sentence does not make sense (grammatically).

Reply: We have rephrased the sentence (lines 305-307).

Section 2.4: For all of these models, please write somewhere which values are used for the different variables.

Reply: Following the reviewer's comments, we have added the values used for the different variables.

Equations 9&11: Please use a consistent notation for the equilibrium fractionation factor.

Reply: We have changed the notation to "$\alpha_v^l$" in Equation 16 and in lines 319 and 321.

Equation 10: This is specific to HDO. I would either add a second equation for H218O, or make the first equation more general.

Reply: We have added the (new) Equation 15 to represent $\left[H_2{}^{18}O\right]$.

Equation 11: I would cite Craig & Gordon (1965).

Reply: Done (line 330).

L351: For the water isotopes the seasonal cycle is not very obvious from Figure 2. The hourly or daily variability is much larger than the seasonality.

Reply: Following the reviewer's comment, we have rechecked the data. From Fig. S3,

there are seasonal variations in relative humidity, specific humidity, lifting condensation level, monthly precipitation, and water vapor isotopic composition ($\delta^{18}O$, $\delta D$, and d-excess).

L352: How did you get the LCL? And why is it relevant?

Reply: We have added the calculation steps for LCL in Section 2.1 (lines 204-209) and as new Equations 1-5. To some extent, LCL can reflect precipitation conditions. Therefore, we chose to use LCL for the analysis.

L357: The maximum temperature is much higher, isn't it?

Reply: Yes, it is. Thank you for pointing this out. We have corrected the value for maximum temperature to "33.5 ℃" in line 389.

Figure 2: I don't think it is necessary to show both humidity and specific humidity.

Reply: We have added an explanation for why we show both humidity and specific humidity (line 231-237). We plotted both in Fig. 2 because, due to weather conditions and instrument trouble, the humidity measured by the LGR instrument is missing data for March to April. Additionally, the meteorological variables measured by the AWS are missing data for September to October, leading to some missing specific humidity values calculated from meteorological parameters. This is why we chose to present both variables as they complement each other, providing a clearer picture of humidity changes at Matara.

L414: emerged -> show

Reply: Changed (line 436).

L442: Why do you compare your values to those from Greenland? It is a very distant site.

Reply: Indeed. Following the reviewer's comment, we have changed this to a comparison with Bangalore station, located in southwest India. Bangalore is also a

coastal city near the Arabian Sea. The revised content can be found in lines 461-468.

L466: What do you mean by precipitation leaching?

Reply: Rainfall exerts a certain leaching effect on moisture and influences the mixing process of water vapor, which is why the observed moisture falls between the Rayleigh fractionation line and the isotope mixing line.

L472: "were" missing

Reply: We have modified the sentence to "The measurements substantially deviate from the Rayleigh curve and show a higher depletion than predicted by the Rayleigh model, likely due to the influence of convective processes." (line 507).

L485: "and led…" does not fit here

Reply: Following the reviewer's comments, we have modified the words "and led" to "due to" in line 501.

L497: What do you mean by reversed? The diurnal variations go in the same direction, only the magnitudes are different.

Reply: Following the reviewer's comments, we have changed the sentence in lines 512-514.

L626: It is averaged in space, isn't it?

Reply: Yes, we calculated the average over a 5°×5° spatial area.

L662: Maybe write here that this is now for the simultaneous values.

Reply: Following the reviewer's comments, we have rephrased this in line 678.

L692: The highest? (Fluctuations)

Reply: Thank you for your comment. Yes, it refers to the maximum value in line 789.

Supplement

L21: directedly -> directly

Reply: Corrected (line 21).

L23: of what?

Reply: Following the reviewer's comments, we have added the word "what" after the "of" in line 23.

L30: led -> lead

Reply: Changed (line 30).

Figure S3: Could you also mark the northeast and southeast monsoon months like in

Reply: Changed as requested

Figure 2?

Reply: Thank you for your comment. I am sorry but I did not quite understand what you mean here. In the *Supporting Information*, there is no Fig. 2.

Figure S4: Maybe add titles to the subfigures to make it clear which is which.

Reply: Following the reviewer's comment, we have added titles.

Figure S5: I don't see the yellow solid line. The figure resolution is not good.

Reply: Following the reviewer's comment, we have rechecked the figure. As a result, we found that the yellow solid line is a misuse and is not included in the image. Therefore, we have removed the "yellow solid line" section (lines 137-139 in *Supporting information*). Meanwhile, we have redrawn the image and increased its resolution.

Figure S7: Again, titles would help to know which subfigure corresponds to which

season.

Reply: Done as requested.

What is the difference between Figures S7 and S9?

Reply: The difference between Fig. S7 and Fig. S9 lies in the time periods. Fig. S7 represents the southwest and northeast monsoon periods, while Fig. S9 represents the whole year.

Figure S10: Why do you show only the northeast monsoon?

Reply: The northeast monsoon is discussed separately because the changes during this period are more distinct and representative.

Table S1: Since VSMOW is there, add SLAP?

Reply: Following the reviewer's suggestion, we have added "VSMOW-SLAP" and "Vienna Standard Mean Ocean Water- Standard Light Antarctic Precipitation" in Table S1.

References

Coplen, T. B. (1994). Reporting of stable hydrogen, carbon, and oxygen isotopic abundances (technical report). Pure and applied chemistry, 66(2), 273-276.

Craig, H., & Gordon, L. I. (1965). Deuterium and oxygen 18 variations in the ocean and the marine atmosphere. In Stable isotopes in oceanographic studies and paleo-temperatures (pp. 9–130). Lab. Geol. Nucl.

Keune, J., Schumacher, D. L., & Miralles, D. G. (2022). A unified framework to estimate the origins of atmospheric moisture and heat using Lagrangian models. Geoscientific Model Development, 15(5), 1875-1898.

Sodemann, H., Schwierz, C., & Wernli, H. (2008). Interannual variability of Greenland winter precipitation sources: Lagrangian moisture diagnostic and North Atlantic Oscillation influence. Journal of Geophysical Research: Atmospheres, 113(D3).

Tuinenburg, O. A., & Staal, A. (2020). Tracking the global flows of atmospheric

moisture and associated uncertainties. Hydrology and Earth System Sciences, 24(5), 2419-2435.

---

## Author Response (AR2)

**response_to_editor**

I have received two reviews on your revised version of this paper. Both reviewers point out the need for additional major revisions to the text for better clarity and to enhance the significance of the paper through an in-depth discussion of the results. In the current form, I cannot accept this paper, and I encourage you to take the additional comments as well as the remarks below on the conclusions very seriously and carefully and thoroughly implement the requested changes.

Reply: Thank you for your suggestion. We have made reply and modification item by item according to your comments, and the line number in the reply is the line number of the clean version.

In addition to the reviewers comments please carefully revise the writing in the abstract and conclusions:

- L. 28 and 32: lower and higher than what? Rewrite to "small amplitude and large amplitude" if you do not mention to what you compare these amplitudes in the same sentence.

Reply: Thank you for your suggestion. We have modified the sentence from "lower and higher" to "small and large" (lines 27-30).

- L. 813: "deficiency of water vapour isotope values" what do you mean by deficiency? That they are not useful? Or do you mean depleted water vapour isotope values?

Reply: Thank you for your suggestion. We have modified the word from "deficiency" to "depleted" (line 774) as follows.

The depleted of water vapor isotope values at Matara station in autumn and winter is consistent with findings from other coastal stations, such as Bangalore, Ponmudi, and Wayanad (Rahul et al., 2016b; Lekshmy et al., 2018).

- L. 816: here you need to make clear what you mean by OLR at 1-4 days. 1-4 days locally or along the trajectories? 1-4 days before the observation?

Reply: Thank you for your suggestion. According to the suggestion of reviewer 1, we

replaced the OLR datasets (source with ERA5) and recalculated the spatial distribution of the correlation between water vapor stable isotopes and OLR. The results show that the maximum spatial correlation coefficients occur 2–5 days before the corresponding day (n=0).

2-5 days means local and before the observation. The 2-5 days denote the period over which the spatial correlation is assessed between daily averaged in-situ $\delta^{18}O$ measurements of water vapor and OLR, starting from the current day (n=0) to the preceding n days (lines 637-639 and 765-767).

- In the conclusions I still miss an in-depth discussion of the results in view of the existing literature. I mentioned this in the very beginning of the review phase and, unfortunately, I see only now that the conclusions do still not comply with the ACP guidelines:https://www.atmospheric-chemistry-and-physics.net/policies/guidelines_for_authors.html

Most importantly, I miss a "Comparison and context": there are other publications that have been looking at the impact of convection on the isotope composition in subtropical regions, please take this literature into account and mention what the new findings are with respect to the existing literature (Bailey et al. 2015, Risi et al. 2019, Benetti et al. 2015, De Vries et al. 2022, Galewsky et al. 2023, Landshuter et al. 2024). What do the isotope signals in Matara reveal, that others have shown using conceptual models or numerical model simulations?

Furthermore, Caveats and limitations are not discussed. This is an important part of the conclusions. Please add a paragraph on the limitations of your analysis. For example, you base your interpretations on a single year of observations, and the trajectory-based analysis lacks a mass-based budgeting approach.

Finally, the implications are a bit far away from the results presented, please bridge to the last paragraph by mentioning how concretely these observations may be used in the future to address open questions about water cycle changes in this region.

Reply: Thank you for your suggestion. We have rewritten the conclusion in Section 5 (lines 735-810).

Following your suggestions, we have added the "Comparison and context" in the third paragraph in *Section 5 (Summary and Conclusion)* (line 779-791) and "Caveats and limitations" in the fourth paragraph *Section 5 (Summary and Conclusion)* (lines 766-768 and 801-803). We described the application of the observation results in solving the water cycle changes in the region (lines 792-797).

Furthermore, we also added the description of water vapor stable isotope dataset (lines 739-743) and the correlation analysis of moisture sources based on HYSPLIT (lines 753-759).

In my view, the additional requests for major changes to the text by the two reviewers and myself are clear enough for you to implement the necessary changes. Please take the time to carry out these implementations carefully.

**response_to_reviewer1**

The new version of the manuscript is clearer, but not all reviewer comments have been properly addressed in my opinion. I suggest another round of minor revisions before the manuscript can be published.

General comments

1) I still don't understand how moisture sources are defined. Did you look at changes in specific humidity along the trajectories and grid them? If so, did you do any discounting (i.e. if moisture is lost later, does an increase in specific humidity still fully count as a moisture source)? Or did you define the end points of the trajectories as moisture sources? Please describe this clearly.

Reply: Thank you for your suggestion. To elucidate the spatial distribution of near-surface water vapor sources at the Matara station, Lagrangian backward trajectory simulations were conducted. All backward trajectories arriving at the station during the southwest and northeast monsoons were spatially clustered, with each cluster analyzed for temporal changes in parameters such as air parcel altitude and specific humidity. This approach enabled the identification of the air mass origins. Fig. 5(a, b) depict the spatial clustering of backward trajectories and the changes in specific humidity along these trajectories for the southwest and northeast monsoons, which are all gridded.

We have not done any discounting. The end points of trajectories are indicative of the moisture sources. The text has been modified in this paper (lines 530-534) and Fig. S6 has been added in *Supporting Information*.

2) The introduction is more streamlined now, but it still goes back and forth a bit. For example, I wouldn't mention isotopes in the paragraph starting at L68 yet; on L109 there is a jump from isotopes back to water cycle and back to isotopes; the paragraph starting at L136 should probably come somewhere in the beginning of the isotope part, because it is a general motivation for the use of isotopes.

Reply: Thank you for your suggestion. The introduction has been reorganized to

improve its coherence.

We have deleted the isotope part of the second paragraph (line 70).

The third paragraph (the paragraph starting at L136 in original version) now includes an explanation of the importance of water vapor stable isotope research (lines 79-84). Furthermore, the section on the influence of sea surface evaporation on the water cycle, which originally preceded the fourth paragraph, has been omitted to create a better connection between the third and fourth paragraphs (line 113).

3) Having a discussion is a good idea, but the first section reads more like another results section than a discussion. The idea of a discussion is to set the results into a broader context. The second discussion section achieves this quite well, but the first section could be (re)moved.

Reply: Thank you for your suggestion. We reorganized the discussion section by transferring the content of the original Section 4.1 to Section 3.3 (lines 559-583). As a result, the discussion section exclusively contains the material from the former Section 4.2.

Specific comments

L28: Better "fluctuations" instead of values (also on L30)

Reply: We have modified the word (lines 28 and 30).

L37: I would change to "Furthermore, the new dataset will enable ..."

Reply: We have modified the sentence (lines 38-39).

L114 (and others): Do you mean kinetic fractionation?

Reply: We have modified the words from "dynamic fractionation" to "kinetic fractionation" (lines 116, 131, and 593).

L131: I would cite Merlivat & Jouzel (1979) instead of Bonne et al. (2019)

Reply: We have modified the reference from "(Bonne et al., 2019)" to "Merlivat and

Jouzel, 1979" (lines 133).

L211: I would move this to after ERA5 variables have been described.

Reply: Thank you for your suggestion. We have moved this text to the section describing ERA5 (lines 211-215).

L225: Since ERA5 provides OLR, I would use that instead of OLR from NCEP, for consistency with the other variables.

Reply: Thank you for your suggestion. We use OLR datasets obtained from ERA5 to analyze the impact of regional convective activity on atmospheric water vapor stable isotopes (lines 209-211) and redraw Fig. 2 (line 404) and Fig. 8 (line 678).

L232: Can you convert one of them and show both lines in the same plot (either humidity or specific humidity)?

Reply: Thank you for your suggestion. We have plotted the specific humidity obtained the AWS and ERA5 in Fig. S3 (lines 119-123 in *Supporting Information*).

L243+: I assume you mean T_2m air here as well?

Reply: Yes. This refers to the 2m air temperature obtained from the ERA5 datasets (lines 207-211). Therefore, we have modified the formula from "$T_{air}$" to "$T_{2m\ air}$" (lines 229 and 232 and Equation 7).

L244: Shouldn't the right hand side of the equation be q_s(sea surface salinity of 0 PSU)?

Reply: According to Curry and Webster (1998), $q_{sat}$ (SST) is the saturation specific humidity at a sea surface salinity of 35 PSU. We have explained in the parameter description below the Equations 7-8 (lines 229-235).

L248: What do you mean by calculated?

Reply: The calculation here refers to Equation 7. What we mean is that the sea surface

pressure value is taken as atmospheric pressure value to participate in the Equation 7 (lines 233-235).

L318: These equations should be normalized by VSMOW.

Reply: Yes. The data used for these formula calculations has been calibrated by VSMOW. We have added the description as follows (lines 307-308).

The isotopic ratio and isotopic δ in the Eq. 14 and Eq. 15 have been calibrated by VSMOW.

L342: And also from higher levels. It's enough to mention the release heights only once (but all of them).

Reply: We have made deletions and adjustments to this paragraph, mentioning the release heights only once (lines 325-331).

L348: This is a repetition from L340.

Reply: We have removed this sentence (lines 327-331).

L375: I still don't find this seasonal cycle very clear. I would choose a different wording.

Reply: We have modified the description (lines 357-359) as follows.

We demonstrated the monthly variations of the average relative humidity, specific humidity, monthly precipitation, and water vapor isotopic composition ($\delta^{18}O$, $\delta D$, and d-excess) (Fig. S3 and Table 1).

L403: "Consequently": This is a consequence of what?

Reply: We have modified the word to "Therefore" (line 387).

L415: How do you know that the high values of d-excess are related to moisture recycling?

Reply: This sentence has been deleted in the revised version.

L433: "n" missing.

Reply: We have corrected the word of "in" (line 412).

L461: Remove "is"

Reply: We have removed "is" (line 447).

L482: I don't think "leaching" is the correct term. Is this really what you mean? https://en.wikipedia.org/wiki/Leaching_(chemistry)

Reply: Thank you for your suggestion. The word has been deleted in the revised version (line 468).

L496: A verb is missing in this sentence.

Reply: We have modified the sentence (lines 481-485).

L523: Where is Lena station?

Reply: The Lena River station is located in eastern Siberia. We have indicated the specific location in the paper (lines 508-510).

L584: Marara -> Matara

Reply: We have modified the word from "Marara" to "Matara" (line 597).

L587: How did you define these regions?

Reply: We have made the relationship between the near surface water vapor d-excess at Matara station and the relative humidity of the surrounding sea area during the observation period (Fig. S9). The water vapor d-excess has a significant negative correlation with the relative humidity in the BoB and the northern Indian Ocean. This is consistent with the trajectory tracking results (Fig. 5), which proves that the water vapor at Matara station is mainly supplied by the surrounding ocean. Therefore, we selected "Region a" and "Region b" (Fig. S9).

L764: 2040 mm per year?

Reply: Yes. We have added the words of "per year" in this sentence (line 714).

L780: Better "that in general agree with..."

Reply: We have modified the sentence as follows (line 730).

We could also identify seasonal patterns that in general agree with previous findings for tropical equatorial regions (Midhun et al., 2013; Rahul et al., 2016b; Lekshmy et al., 2018).

**response_to_reviewer2**

The authors show one year of near-surface water vapor stable isotope data observed in Sri Lanka, which is undoubtedly very important. However, there are still some parts of the paper that need to be improved.

Specific comments:

Line 79: Why mention "river water"? It's not relevant to the topic of this paper.

Reply: Thank you for your suggestion. We have moved the words of "river water" (line 84).

Line 202: It should be added why the LCL is calculated; another reviewer similarly raised this issue, but the author's response was not clear. In the latest version of the paper, it is still not clear that there is a need to discuss the LCL, and there is little connection to the topic of the paper.

Reply: Thank you for your suggestion. We have moved the part of LCL. Accordingly, we modified Table 1 (line 423), Fig. 4 (line 511), and Fig. S3 (line 119 in *Supporting Information*).

Line 381: This Celsius unit has a different font than the others.

Reply: We have corrected it (line 365).

Lines 401-403: It is mentioned that δ18O values are higher during the Southwest Monsoon, but Lines 557-558 mention that δ18O values are lower during the Southwest Monsoon, so why are the results opposite?

Reply: Thank you for your suggestion. We are so sorry for the mistake caused by previous negligence. At present, we have carefully examined the original data, modified the sentence (lines 545-546), and revised the Fig. 5.

Lines 404-405, Lines 410-411: "δ18O decreases during ... periods"; "d-excess increases during ... periods". These statements are confusing, and no decrease or

increase trend can be seen from Figure 2.

Reply: According to Table 1, we revised these two sentences as follows, please see lines 388-390 and lines 394-396.

During the southwest monsoon, the northeast monsoon, and the non-monsoon periods, the average values of $\delta^{18}O$ are -11.1‰, -12.2‰, and -11.9‰, respectively.

Furthermore, during the northeast monsoon, the southwest monsoon, and the non-monsoon periods, the average values of d-excess are 12.4‰, 13‰, and 14.7‰, respectively.

Line 414: Line 412 mentions that the maximum d-excess value occurs in November; what is the logic of singling out April here? Is the reason for both November and April d-excess values being high the same? Are they both caused by local recycling?

Reply: Thank you for your suggestion. We have moved the sentence of "d-excess peaks in April 2020 at 19.1‰, indicating potential contributions from local recycling." We determined that the higher d-excess in November was due to local recycling.

Figure 2: The word 'pressure' is incomplete.

Reply: Thank you for your suggestion. It is corrected in Fig. 2 (line 404).

Line 438: "$\delta$" → "$\delta$18O"

Reply: We have modified the word from "$\delta$" to "$\delta^{18}O$" (line 417).

Lines 436-438: "Over the 75-day period spanning from … between -22‰ and -11‰." The beginning of the paragraph mentions that there were abrupt changes in stable isotopes during this period, related to synoptic events. Here it should be summarized exactly what events are associated with it, rather than simply describing the isotopic signature.

Reply: Thank you for your suggestion. A single weather event is discussed in lines 412-415 and 417-419.

Lines 438-443: "During the southwest monsoon from July 12 to August 7…below the minimum in the Bay of Bengal (Midhun et al., 2013)." Why do you single out the period from July 12 to August 7 for analysis of isotopic changes? Is there anything special about this period? Is it related to the abrupt changes or synoptic events mentioned at the beginning?

Reply: From July 12 to August 7, there was a significant oscillation and fluctuation in the water vapor stable isotopes at Matara station, with a sharp depletion of $\delta^{18}O$ (Fig. 2). Corresponding to heavy precipitation events during the same period. Therefore, it will be described here as a weather event and related to the synoptic events mentioned at the beginning.

Lines 443-445: "Other coastal stations such as Bangalore, Ponmudi, and Wayanad also exhibit … observations at Matara (Table 2)." These analyses of the characteristics of autumn or winter changes, not synoptic changes, should be placed in the previous paragraph.

Reply: Following your suggestion, we have removed this sentence to the end of the third paragraph in Section 3.1 (lines 401-403).

Lines 454-457: Regarding the interpretation of the slope, it seems to be different from previous studies, which suggested that lower slopes are usually associated with evaporation.

Reply: Thank you for your suggestion. We checked the previous statement and found some errors. Therefore, we reanalyzed the causes underlying the variability in the slope and intercept of LMWL during the southwest monsoon and northeast monsoon as follows (lines 437-443).

The LMWL slope and intercept vary significantly between monsoon and non-monsoon seasons, peaking in the northeast monsoon with values of 7.3 and 3.86, and nadir in the southwest monsoon with 6.93 and 1.18, respectively. This suggests increased humidity over sea surface vapor sources from the northeast to southwest monsoon, attributed to heightened evaporation and reduced dynamic fractionation

effects. During the northeast monsoon, LMWL slope and intercept are higher compared to other periods, indicating significant moisture recirculation.

Lines 490-491: In addition to convective processes, is the effect of raindrop re-evaporation considered? In addition, is it possible to demonstrate that convective activity is stronger during the northeast monsoon through the spatial distribution of the OLR? Finally, Section 4.1, lines 716-717 mentions that the air brought by the northeast wind is drier during the northeast monsoon. Does dry air contribute to the enhancement of convection?

Reply: This article only considers the convective process and does not take into account the impact of raindrop re evaporation. According to the spatial distribution of the correlation between OLR and $\delta^{18}O$, the results reveal that the correlation is significantly stronger during the northeast monsoon, which indicates that convective activity is stronger during the northeast monsoon period compared to the southwest monsoon period. However, the Matara station in Sri Lanka remains moist throughout the year, with a specific humidity range of 17-22 g/kg.

Our work does not delve deeper into the potential role of dry air in enhancing convection. We sincerely acknowledge the reviewer's insightful feedback and plan to incorporate this aspect into our future investigations.

Line 587: Why not choose the region with stronger negative correlation within the 50-60 °E range in Figure S6 for "Region a"?

Reply: Thank you for the reviewers' suggestions. We modified the selection of regions and selected regions with stronger negative correlation in Fig. S9. We have calculated the values of $RH_{SST}$ and the relationship between the values of $RH_{SST}$ and d-excess during the southwest monsoon. The calculation results (Fig. 7) and the expression (lines 596-621) in the paper are modified.

Lines 636-638: In Figure 7, the correlations between δ18O and precipitation, OLR during the northeast monsoon are also very high on the west of the study site, which is

inconsistent with the results of the moisture trajectory (from the northeast), why?

Reply: Thank you for the reviewers' suggestions. We have replaced the OLR dataset from ERA5. And the spatial distribution of correlation coefficients between $\delta^{18}O$ and OLR was recalculated. The results of the correlation during the northeast monsoon are consistent with the moisture trajectory in the revised version (Fig. 5 and Fig. 7).

Line 643: There are only temporal correlation results in Figures 7e and f. Where are the spatial correlation results? In addition, a 5×5 region could be presented in the figure, which would be more intuitive.

Reply: Thank you for your suggestion. At present, after adjustment, Fig. 7 is changed to Fig. 8. Figures 8a-d show the temporal and spatial correlation, while Figures 8e-f only show temporal correlation. The wording in the text has been revised (lines 654-656). In addition, a 5×5 region has been presented in Fig. 8 (lines 678-689).

Lines 677-678: Section 3.4 is titled Influence of Convective Activity, but the last two paragraphs discuss the relationship between stable isotopes and temperature and relative humidity, respectively, which is not relevant to the topic and does not add valid information, so it is recommended that it be deleted.

Reply: Following your suggestion, we have removed the last two paragraphs in *Section 3.4* and the related figures in *Supporting Information*.

Lines 739-744: Moisture from the northeast during the southwest monsoon? During the northeast monsoon, the moisture comes from the southwest? It's the opposite, isn't it?

Reply: Following your suggestion, we have rewritten these sentences (lines 569-580).

The headings of Sections 3.2, 3.4, 4.1 and 4.2 are not specific and it is recommended that they be revised.

Reply: Following your suggestion, we modified the heading of Section 3.2 to "The Variation Characteristics of Diurnal Cycles" (line 477), the heading of Section 3.4 to

"The Influence of Regional Convective Activity" (line 628), and the heading of Section 4 to "Discussion: Comparing Main Features and Identifying Influencing Factors" (lines 691-692), respectively.

---

## Author Response (AR4)

**response_to_editor**

All reviewer and editor comments were addressed in a satisfactory way. I am happy to recommend this paper for publication in ACP. One last change necessary change is related to the title: it should follow the ACP author guidelines: https://www.atmospheric-chemistry-and-physics.net/policies/guidelines_for_authors.html

"Titles should be concise and consistent with the content and purpose of the article. For research articles, ACP prefers titles that highlight the scientific results/findings or implications of the study."

In the current form, the title rather sounds like a technical report or a data publication, please change the title such as to include the main scientific finding of your study.

E.g. something along the line of: "One-year Continuous Observations of Near-Surface Atmospheric Water Vapor Stable Isotopes at Matara in Sri Lanka reveal a strong link to the intensity of convection".

Reply: Thank you for your suggestion. We have revised the title to "One-Year Continuous Observations of Near-Surface Atmospheric Water Vapor Stable Isotopes in Matara, Sri Lanka Reveal a Strong Link to Moisture Sources and Convective Intensity".